# Ultra-stretchable and biodegradable elastomers for soft, transient electronics

Won Bae Han [1,7], Gwan-Jin Ko [1,7], Kang-Gon Lee[2], Donghak Kim[3], Joong Hoon Lee [1], Seung Min Yang [1,4], Dong-Je Kim[1], Jeong-Woong Shin [1], Tae-Min Jang [1], Sungkeun Han[1], Honglei Zhou[5], Heeseok Kang [1], Jun Hyeon Lim[1], Kaveti Rajaram [1], Huanyu Cheng [5], Yong-Doo Park[2], Soo Hyun Kim[3] & Suk-Won Hwang [1,3,6] ✉

As rubber-like elastomers have led to scientific breakthroughs in soft, stretchable characteristics-based wearable, implantable electronic devices or relevant research fields, developments of degradable elastomers with comparable mechanical properties could bring similar technological innovations in transient, bioresorbable electronics or expansion into unexplored areas. Here, we introduce ultra-stretchable, biodegradable elastomers capable of stretching up to ~1600% with outstanding properties in toughness, tear-tolerance, and storage stability, all of which are validated by comprehensive mechanical and biochemical studies. The facile formation of thin films enables the integration of almost any type of electronic device with tunable, suitable adhesive strengths. Conductive elastomers tolerant/sensitive to mechanical deformations highlight possibilities for versatile monitoring/sensing components, particularly the strain-tolerant composites retain high levels of conductivities even under tensile strains of ~550%. Demonstrations of soft electronic grippers and transient, suture-free cardiac jackets could be the cornerstone for sophisticated, multifunctional biodegradable electronics in the fields of soft robots and biomedical implants.

The unique feature of transient technology, which dissolves or decomposes into biologically benign end products in certain conditions after completion of functional operations, provides new opportunities to develop a wide range of research areas, including temporal biomedical implants[1–6], green and nature-friendly electronic devices[7–9], and electronic security systems[10–12]. However, further developments in diverse, sophisticated functionality or system have been constrained by the dependence on a narrow range of options or characteristics of biodegradable materials, particularly substrates that play key roles in the physical, electrical, and chemical properties of the whole systems. Early flexible and stretchable electronics, which were developed based on silicone elastomers, polydimethylsiloxane (PDMS), and Ecoflex, have been expanded into wearable and implantable electronic systems with numerous applications, ranging from electronic skin (e-skin)[13,14], soft robotic machine[15,16], and epidermal virtual reality[17], to brain- or cardiac-interfaced devices[18,19] and other electronic components and

[1]KU-KIST Graduate School of Converging Science and Technology, Korea University, 145 Anam-ro, Seongbuk-gu, Seoul 02841, Republic of Korea. [2]Department of Biomedical Sciences, College of Medicine, Korea University, 145 Anam-ro, Seongbuk-gu, Seoul 02841, Republic of Korea. [3]Center for Biomaterials, Biomedical Research Institute, Korea Institute of Science and Technology (KIST), 5 Hwarang-ro 14-gil, Seongbuk-gu, Seoul 02792, Republic of Korea. [4]Hanwha Systems Co., Ltd., 188 Pangyoyeok-ro, Bundang-gu, Seongnam-Si, Gyeonggi-do 13524, Republic of Korea. [5]Department of Engineering Science and Mechanics, The Pennsylvania State University, University Park, PA 16802, USA. [6]Department of Integrative Energy Engineering, Korea University, 145 Anam-ro, Seongbuk-gu, Seoul 02841, Republic of Korea. [7]These authors contributed equally: Won Bae Han, Gwan-Jin Ko. ✉e-mail: dupong76@korea.ac.kr

systems[20,21]. In this aspect, researches on various new biodegradable substrates are required. Besides inelastic degradable materials, such as synthetic and natural polymers, recent research efforts have introduced a few different types of biodegradable and stretchable materials based on protein[22], hydrogels[23–25], and crosslinked polymers[26–29]. However, most of them are far from practical and multipurpose uses, unlike their non-transient counterparts, due to insufficient characteristics, such as uncontrollable dissolution behaviors[23–25,27,29,30], limited mechanical elasticity/modulus/vulnerability[22–24,26,28,30–32], or poor storage stability[22,24,25,29,30] (Supplementary Table 1).

In the following, we introduce a super-elastic and biodegradable elastomer, poly(l-lactide-co-ε-caprolactone) (PLCL), and re-discover the values in unexplored forms beyond conventional, restricted uses as porous scaffolds[33,34] and a few implementations as a substrate for electronic materials[35–37]. The elastomer is readily applicable to existing microfabrication processes, suitable for substrates and encapsulants of transient, deformable electronics. Elastomeric composites incorporated with organic/inorganic conductive elements produce dissolvable electronic circuits, electrophysiological (EP) monitors, and extremely sensitive arrays of pressure sensors. System-level demonstrations include smart, soft robotic grippers combined with electronics that can perceive the external circumstances (e.g., temperature and pressure) and transient, suture-free cardiac jackets with bio-inspired designs for the management and treatment of cardiac functions over long periods of time.

## Results

### Stretchable, biodegradable PLCL elastomers with versatile functions

Figure 1a shows a brief description of chemical synthesis to produce a biodegradable, ultra-stretchable elastomer, PLCL, by coupling of cyclic esters of L-lactide (LA) and ε-caprolactone (CL) through ring-opening polymerization, where physical cross-linking of hard (LA) and soft domains (CL) in a 5:5 ratio provides higher polymer elasticity and softness than previous works using high LA fractions[38,39] (Supplementary Fig. 1). Varying monomer to initiator ratios yields different molecular weights ($M_n$) of PLCL elastomers (Supplementary Table 2), allowing for tunable chemical/physical/mechanical properties. Figure 1b highlights various features of PLCLs engineered with a flower shape, including superior stretchability (~1600%) and enzymatic/chemical degradability, applicable for various research fields of transient, biodegradable electronic systems. Examples functions and areas of interest involve (1) substrates and encapsulants in ultrathin geometries for degradable electronics, (2) polymer matrices for conductive elastomer composites, (3) three-dimensional (3D) free-form frameworks for soft robotics, and (4) elaborate design layouts for soft, suture-free medical implants.

For the evaluation of biological degradation in the natural environment as a zero environmental footprint, we conducted experiments using *Aspergillus flavus* under aerobic conditions (relative humidity (RH), 50%; temperature, 37 °C) (Fig. 1c). The result exhibits that microbial digestion continuously decomposed PLCL films over time via gradual breakdown of complex polymers into biologically benign fragments[40] (Fig. 1d and Supplementary Fig. 2), as the linear reduction of weight ratio to ~40% after 30 days (Fig. 1e).

### Biochemical/mechanical/physical characteristics of PLCL elastomers

Similar investigations, but in water or biofluids, are also important. Figure 2a shows a set of dissolution images of PLCL, collected at periodic timescales when half-immersed in a phosphate buffer solution (PBS, pH 7) at body temperature (37 °C). PLCL films (140k) dissolved via hydrolysis over time without remaining any toxic byproducts[41,42] and completely disappeared at 30 weeks, while dissolution behaviors can be varied with types, temperatures, and pH levels of solutions

(Supplementary Fig. 3). Inspection on different molecular weights in Fig. 2b indicates expected outcomes that degradation rates became faster as the molecular weight of PLCL decreased, since the higher molecular weight, the more polymer chain cleavages required to reach a critical molecular weight at which the polymer begins to dissolve[40]. The overall dissolution behaviors of PLCL for all factors studied here are consistent with those of materials discussed in previously published articles[40,43]. Figure 2c provides an extremely elastic feature that could accommodate linear strains up to ~1600% (Other severely deformed demonstrations appear in Supplementary Fig. 4). Here, different molecular weights offered a variety of options for modulating mechanical characteristics, including tensile strengths ($\sigma_u$, 5–20 Mpa) and elongations at break ($\varepsilon_b$, 700–1600%) (Fig. 2d), with high recovery rates (70–90%) against uniaxial strains (600–1000%) (Supplementary Fig. 5). In particular, cyclic tests of PLCL (200k) at a uniaxial strain of 40% exhibited negligible elastic hysteresis (Fig. 2e), which outperforms similar types of elastomers introduced in recent reports[26,27]. Although the cyclic tests show hysteresis of PLCL, particularly at high strains, further studies need to improve the property through various chemistry. Furthermore, excellent toughness (35–45 MJ/m$^3$) and crack-tolerance (30–140 kJ/m$^2$) that are much higher than those of other biodegradable polymers and silicone elastomer, PDMS (fracture energy, 0.35 kJ/m$^2$)[44] (Fig. 2f), enabled PLCL elastomers with a cut notch to elongate over ~300% without fracture via energy absorption (Supplementary Fig. 6). The extent of variations in mechanical properties, which may occur during partial dissolution or storage, can be an important criterion for effective functional timescales. Figure 2g shows gradual changes in properties of $\sigma_u$ and $\varepsilon_b$ during immersion in PBS (pH 7) at 37 °C. Each characteristic was maintained more than 60% ($\sigma_u$) and 90% ($\varepsilon_b$) of the intrinsic performances on the dissolution of 20%, while these mechanical properties remained nearly constant in ambient conditions during 6 months (Fig. 2h). Additional water swelling ratio appears in Supplementary Fig. 7. Compared with the properties of well-known biopolymers such as silk[45], poly(lactic acid) (PLA)[46], poly(lactic-co-glycolic acid) (PLGA)[47], and polycaprolactone (PCL)[48], and recent degradable elastomers, such as biogel[23], TRn25[22], b-DCPU70[26], PseD-U[27], PGSA-19[28], and poly(1,8-octanediol-co-citric acid) (POC)[49] (Fig. 2i), the exceptional range of stretchability of PLCL elastomers suggests the possibility as a promising candidate for a wide range of transient, stretchable systems. Figure 2j shows a partially degradable Arduino-powered electronic display with an array (8×16) of commercially available, non-degradable red, green, and blue (RGB) light-emitting diodes (LEDs) on dissolvable PLCL substrates manufactured by microlithography and transfer-printing processes. Along with resilience, high optical transparency (~90%), frequency-independent permittivity (dielectric constant, ~3.5), and appropriate adhesion force facilitated a reliable, stable operation under various deformation modes of folding, poking, and bending (Supplementary Fig. 8 and Supplementary Movie 1). Integration with bioresorbable silicon-based components produced fully dissolvable, sophisticated integrated circuits, e.g., complementary metal-oxide-semiconductor (CMOS) inverter arrays, whose stretchability reached up to ~200% (Supplementary Fig. 9). Superior resistance to thermal and chemical factors enables direct fabrication procedures onto the elastomeric substrates, which was realized by patterned microheater arrays with sufficient capacity (Supplementary Fig. 10).

### PLCL-based conductive elastomeric composites for bio-integrated electronics

Conductive elastomers offer opportunities for flexible electronics and soft robotics over conventional inorganic-based metal electrodes. Figure 3a illustrates partially biodegradable, conductive elastomeric composite composed of PLCLs as a polymer matrix, poly(3,4-ethylenedioxythiophene):poly(styrenesulfonate) (PEDOT:PSS) as a biocompatible, conductive filler[50], and N-methyl-N-butylpyrrolidinium

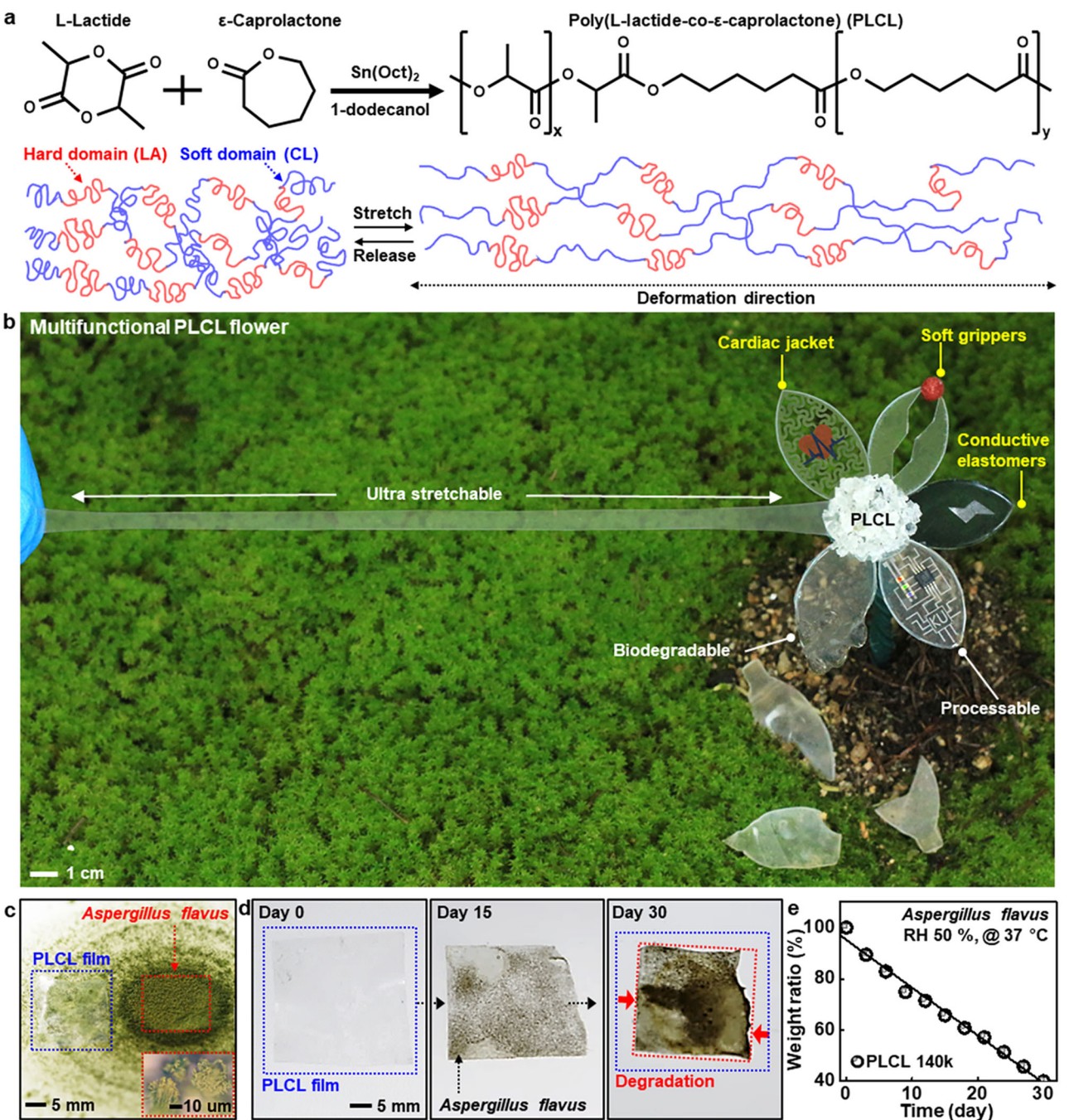

**Fig. 1 | Highly stretchable, processable, biodegradable elastomers for versatile applications of transient electronics. a** Molecular structure of poly(L-lactide-co-ε-caprolactone) (PLCL) elastomer synthesized from biocompatible/biodegradable cyclic esters, L-lactide and ε-caprolactone, and proposed mechanism for reversible chain motions during tensile stretching. **b** PLCL elastomer-based artificial flower, whose individual leaf describes features of stretchability, degradability, and processability, and potential applications for conductive polymer composites, soft robotics, and bio-integrated electronics. **c** A thin PLCL film (thickness, ~100 μm) on fungi *Aspergillus flavus*, for enzymatic degradation, with a magnified view of the fungi in the inset. **d, e** A collection of optical images of the enzymatic degradation (**d**) and the corresponding changes in weight ratio (**e**) of a PLCL film (140k) at 37 °C under relative humidity (RH) of 50%.

bis(trifluoromethanesulfonyl) imide (P14[TFSI]) as a biocompatible/biodegradable plasticizer[51]. The composites (PLCL:PP) with an optimized weight ratio (PLCL:PEDOT:PSS:P14[TFSI], 0.6:0.4:1) exhibited excellent initial conductivity (~220 S/cm) and stretchability (550%), with robust cyclic stability under 40% strain (Supplementary Fig. 11). Such outstanding electrical performance might be originated from the use of dimethylsulfoxide (DMSO) as a co-solvent, which induces phase separation of conductive PEDOT and insulative PSS domains to yield microstructures favorable for electrical conduction[50]. Figure 3b, c present tunable characteristics of electrical/mechanical properties via

control of individual elements. Increases in PEDOT:PSS content enhanced conductivity but reduced stretchability owing to the brittle nature of semicrystalline PEDOT:PSS with a fracture strain of <~5%[52] (Fig. 3b). On the other hand, the addition of ionic liquid (P14[TFSI]) improved the elasticity due to the improved segmental motions in polymer chains[53], revealing the obvious role of the ionic liquid in plasticization as well as electrical connections between PEDOT-rich domains although the conductivity somewhat decreased (Fig. 3c). Comparison results with a nonionic fluorosurfactant (Zonyl FS-300) as a plasticizer appear in Supplementary Fig. 12. Such elastomeric

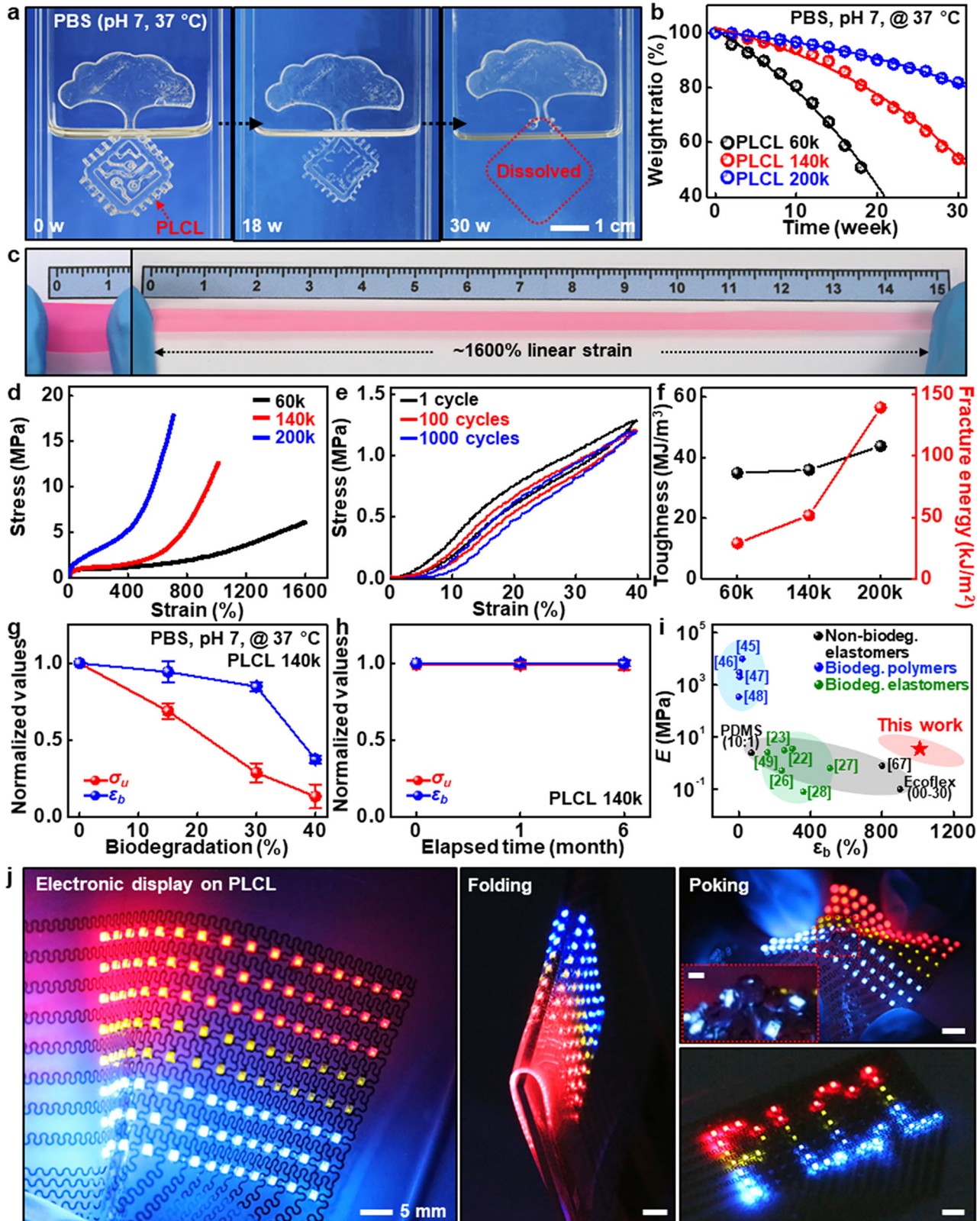

conductors with the excellent mechanical/electrical features enabled the robust operation of electronic circuits built with a LED under several deformation modes such as rolling (R, ~3 mm), folding (~180°), and stretching (~400%) without failures (Fig. 3d and Supplementary Movie 2). Other demonstrations in Fig. 3e and Supplementary Fig. 13 involved measurements of electrocardiogram (ECG) and electromyogram (EMG), whose signal-to-noise ratios were comparable to

those of commercial Ag/AgCl electrodes. Figure 3f shows a set of dissolution images of conductive electrodes while submerged in buffer solution under an accelerated condition (pH 11, 37 °C), and the dissolution rate was slower than that of PLCL itself, presumably due to the water-insoluble PEDOT domains and hydrophobic ionic liquid. Changes in conductivity and microstructure under a similar condition appear in Supplementary Fig. 14. Here, we note that the overall

**Fig. 2 | Chemical, mechanical, and physical characteristics of PLCL elastomers.**
**a** A series of optical images at several dissolution stages of a PLCL film (140k, 100 μm-thick) in phosphate buffer solution (PBS) (pH 7) at the physiological temperature of 37 °C. **b** Time-dependent changes in weight ratios of PLCL films as a function of molecular weight. **c** Super-elastic response of a colored, narrow PLCL elastomer (60k, 100 μm-thick, 20 × 10 mm) with a linear strain of ~1600%. **d** Stress-strain curves of PLCLs with different molecular weights (60k, 140k, and 200k). **e** Cyclic mechanical behaviors of PLCL elastomer (200k) under a uniaxial tensile strain of 40%, showing negligible stress-strain hysteresis. **f** Toughness and fracture energies of PLCLs (60k, 140k, and 200k), indicating the highest toughness and fracture energies among all biodegradable elastomers reported so far. **g, h** Normalized changes in the ultimate tensile strength ($\sigma_u$) and elongation at break ($\varepsilon_b$) of PLCL (140k) as a function of biodegradation (%) (**g**) and storage period in ambient conditions (**h**). Data are presented as mean values ± standard deviation. $n = 3$ independent samples. **i** Comparison of Young's moduli (E) and elongations at break of PLCL elastomers (red) with those of widely used but non-degradable elastomers (black; PDMS, Ecoflex, and styrene-butadiene-styrene (SBS)[64]) and representative biodegradable polymers (blue) and elastomers (green). **j** Array (8×16) of individually addressable red, green, and blue (RGB) light-emitting diodes (LEDs) manufactured on PLCL substrates (left) upon various deformation modes of bending (middle) and poking (top right) and Arduino-powered scrolling LED letters, "PLCL" (bottom right).

performance (conductivity > 200 S/cm; stretchability > 500%) as elastic conductor far surpassed those of degradable composites in previous reports[54–59], which facilitates more robust/reliable function in physiological strain range and may contribute to research applications required for elaborate, sophisticated functions under time-dynamic and arbitrary deformation modes. (Fig. 3g)

Conductive elastomers, on the contrary, can also be formed in a way that is very sensitive to external strains. We selected molybdenum (Mo) flakes due to high electronegativity (~2.16) and large aspect ratio (~40) for effective electron tunneling and low percolation threshold over other degradable conductive fillers. Figure 3h shows strain-dependent characteristics of dissolvable molybdenum (Mo) flakes-based composites (PLCL/Mo) with different ratios. Measured changes in fractional resistances exhibited excessive gauge factors (GFs, 5500–7500) within a range of 13% strains, and negligible hysteresis and particularly the low detection limit (~0.01%) are suitable for extremely small physical/mechanical changes (Supplementary Fig. 15). Figure 3i and Supplementary Fig. 16 present an image and 36-color maps of pressure sensor array that is responsive to external mechanical stimulus, which is able to capture pressure distributions of small, light cotton balls (10 and 20 mg). Other cases, such as recordings of vibrational signals on the vocal cords and pulses on the wrist, appear in Supplementary Fig. 17. Dissolution behaviors of PLCL/Mo composites under the biological condition appear in Supplementary Fig. 18.

## Soft, transient electronic actuators

Key features of PLCLs can provide diverse materials and design options for soft robots or interactive interfaces. Figure 4a describes a transient soft robotic gripper that can perceive physical stimuli, e.g., temperature and pressure. The overall structure was configured into three main parts: (1) PLCL actuators with small pneumatic chambers, (2) integrated electronic components (silicon nanomembrane (Si NM)-based temperature sensor and PLCL/Mo diaphragm-based pressure sensor) for the perception of environmental factors, and (3) 3D-printed PLA gripper head for assembly of actuators and pneumatic line. Detailed information on the gripper and an automated pneumatic control system appears in Supplementary Fig. 19. Self-regulating pneumatic pressures were able to precisely adjust the bending angles of the actuator up to ~190° (left, Fig. 4b and Supplementary Movie 3), which was also theoretically considered through a linear elastic model of 3D finite element analysis (FEA) (Fig. 4b, c). Grasp of a ping-pong ball with sufficient capacity is shown in Supplementary Fig. 20. Cyclic durability tests in Fig. 4d revealed that the actuator endured 10,000 cyclic loads at bending angles of ~60, ~90, and ~180° without mechanical hystereses/failures, even under an aqueous solution (PBS, pH 7, 37 °C) and after extended storage (6 months) (Supplementary Fig. 21). Si NMs-based temperature meters with a temperature coefficient of resistance (0.12/°C) and PLCL/Mo-based pressure sensors with a high gauge factor can detect a wide range of external circumstances, independent of repeated bending motions (Supplementary Fig. 22). Figure 4e, f illustrate operational mechanisms and sequences of soft electronic grippers that can actively recognize and adapt to changes in surrounding parameters. Thresholds were preset at 80° (Fig. 4e) and

100 Pa (Fig. 4f) for individual systems. As each value approached the thresholds, the gripper retreated from a thermal bulb (Fig. 4e) or maintained the constant level of force; otherwise, a cotton ball lost its shape (Fig. 4f). Continuous, real-time measurements of both signals reflected various states of movements of electrically programmable or controllable actuators (Supplementary Movies 4 and 5). Customized operation algorithm and flowchart are shown in Supplementary Fig. 23. Modest modifications to the PLCL elastomers can also produce a soft gripper with two different moduli (e.g., assembly of 60k and 200k) and an ultrathin diaphragm-based artificial muscle (Supplementary Fig. 24).

## Soft, transient, suture-free cardiac jackets for heart diseases

The superb elasticity provides an opportunity to create versatile form factors that can bring beneficial aspects to medical implants. Figure 5a introduces a suture-free cardiac jacket designed to be integrated with the elastic, time-dynamic heart for the management of postoperative cardiac systems. The cardiac jacket comprises (1) an open-mesh body built with electronic components such as PLCL/PP for epicardial ECG recording and electrical stimulation, and PLCL/Mo for local myocardial strain detection, and (2) stingray barb-inspired straps for suture-free implantation that can remove possible cardiac infections/failures caused by sutures[60] (Detailed structure appears in Supplementary Fig. 25). Such sophisticated geometries secured highly durable integration without peeling or repositioning against rapid, intensive cardiac movements (~400 Hz for rats), which enables long-term, reliable electrical functions. For the straps, different angles/dimensions yielded various forces (Supplementary Fig. 26). Measured untying forces (~3 Mpa) of the straps in Fig. 5b were much higher than those produced by abnormal inflation (~200%) of the heart. Figure 5c shows the mechanical moduli of different types of system configurations and epicardial tissues. Compared to a pristine film (~2 MPa), a serpentine mesh exhibited a substantial reduction in the elastic modulus (~50 kPa), resulting in mechanical compliance close to that of an epicardial sheet (~35 kPa). Figure 5d presents a set of images of transient electronics coupled to the heart and corresponding physiological activities through in vivo evaluations using rat models (13 weeks; ~550 g) during 8 weeks. The integration process of the cardiac jacket on a 3D heart model and the device connection strategy appear in Supplementary Figs. 27 and 28. The overall bioelectronic system maintained the initial shape and position in a stable mode, and PLCL/PP electrodes on the right atrium (RA) and left ventricle (LV) recorded epicardial ECGs without deterioration of signal-to-noise ratio (SNR) for an extended period of time (Supplementary Movie 6). Charge storage capacity and long-term stability of PLCL/PP under physiological conditions and additional epicardial and surface ECGs appear in Supplementary Fig. 29. Intermittent electrical pulses (0.1 ms, 5 V) across the wide area of LV could produce ventricular capture (Fig. 5e and Supplementary Fig. 30). Measurements of epicardial ECG revealed apparent pacing spikes with a frequent interval (~420 bpm) and amplified QRS complex in contrast to the sinusoidal rhythm without electrical stimulation (300–320 bpm), which demonstrated the ability to manipulate the rate and rhythm of a heartbeat as a pacemaker. Such long-lasting, invariant

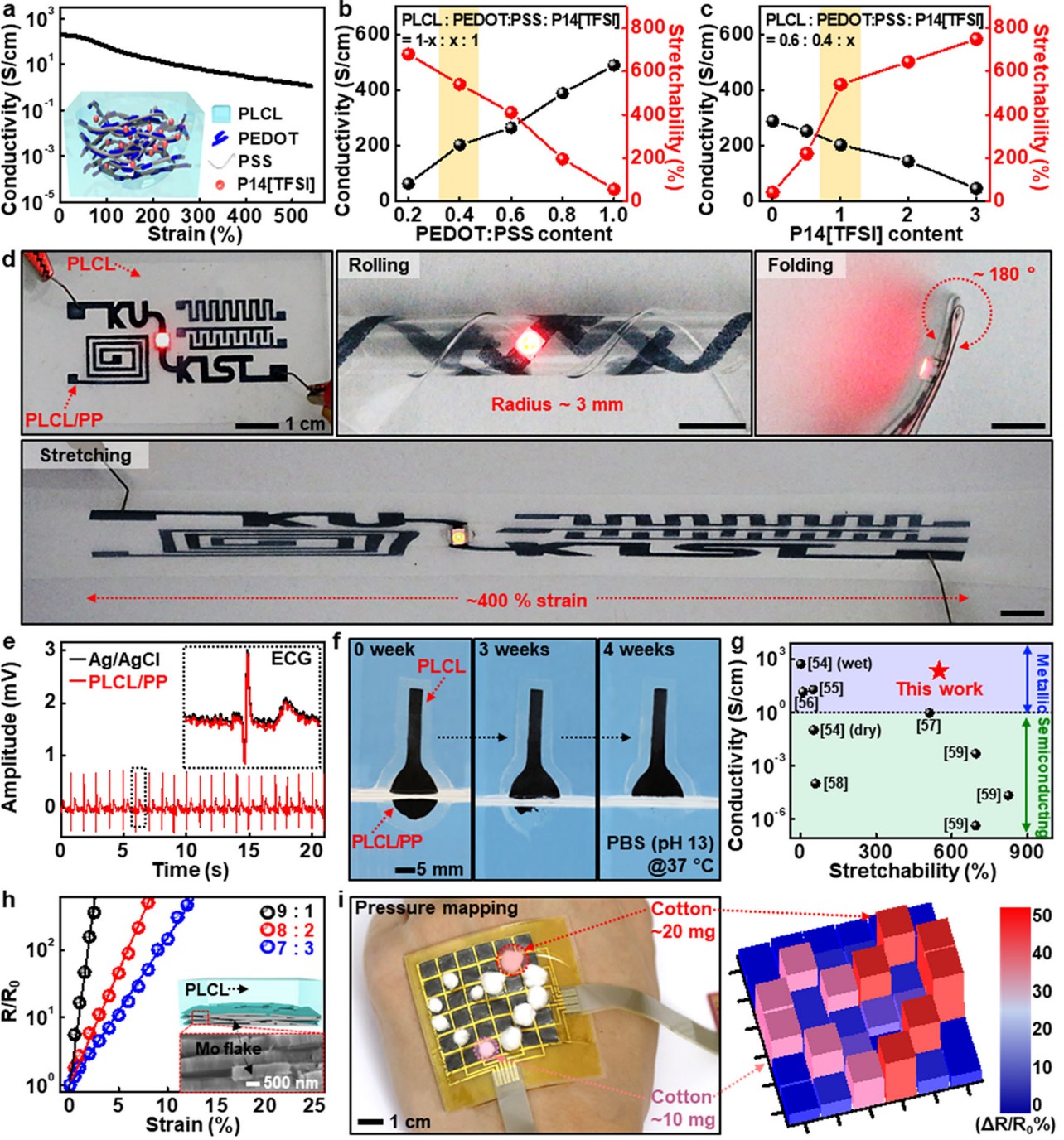

**Fig. 3 | Different types of transient conductive elastomers with organic and inorganic fillers. a** Mechanical strain-dependent electrical characteristics of a partially biodegradable conductive elastomer comprised of PLCL as a polymer matrix, poly(3,4-ethylenedioxythiophene):poly(styrenesulfonate) (PEDOT:PSS) as a conductive filler, and N-methyl-N-butylpyrrolidinium bis(tri-fluoromethanesulfonyl) imide ([P14][TFSI]) as a plasticizer and conductivity enhancer, and a schematic illustration of the PLCL/PEDOT:PSS elastic conductive composite (PLCL/PP) in the inset. **b, c** Dependence of conductivity and stretchability of PLCL/PP on the content of PEDOT:PSS (**b**) and P14[TFSI] (**c**). **d** Examples of partially biodegradable composite (PLCL/PP)-based electronic components with various configurations (top left); demonstrations of reliable and stable operations under various external deformations, rolling (top middle), folding (top right), and stretching (bottom). **e** Measurements of electrocardiogram (ECG) to evaluate electrical performance using PLCL/PP-based probes and commercial Ag/AgCl electrodes. **f** A set of dissolution images of the PLCL/PP-based sensing probe collected over time under an accelerated condition (PBS (pH 13) at 37 °C). **g** Comparison of conductivity and stretchability of the PLCL/PP (0.6:0.4:1) with those of representative biodegradable conductive polymer composites in previous reports. **h** Dissolvable PLCL composite with different amounts of Mo flakes, exhibiting changes in electrical properties that are highly sensitive to strain, and the schematic (top) and enlarged SEM image (bottom) in the inset. **i** Array (6×6) of PLCL/Mo-based pressure-sensitive system with passive matrix addressing (left) and measured distribution of pressures generated from very light weights of cotton balls (-10 mg, 20 mg) (right).

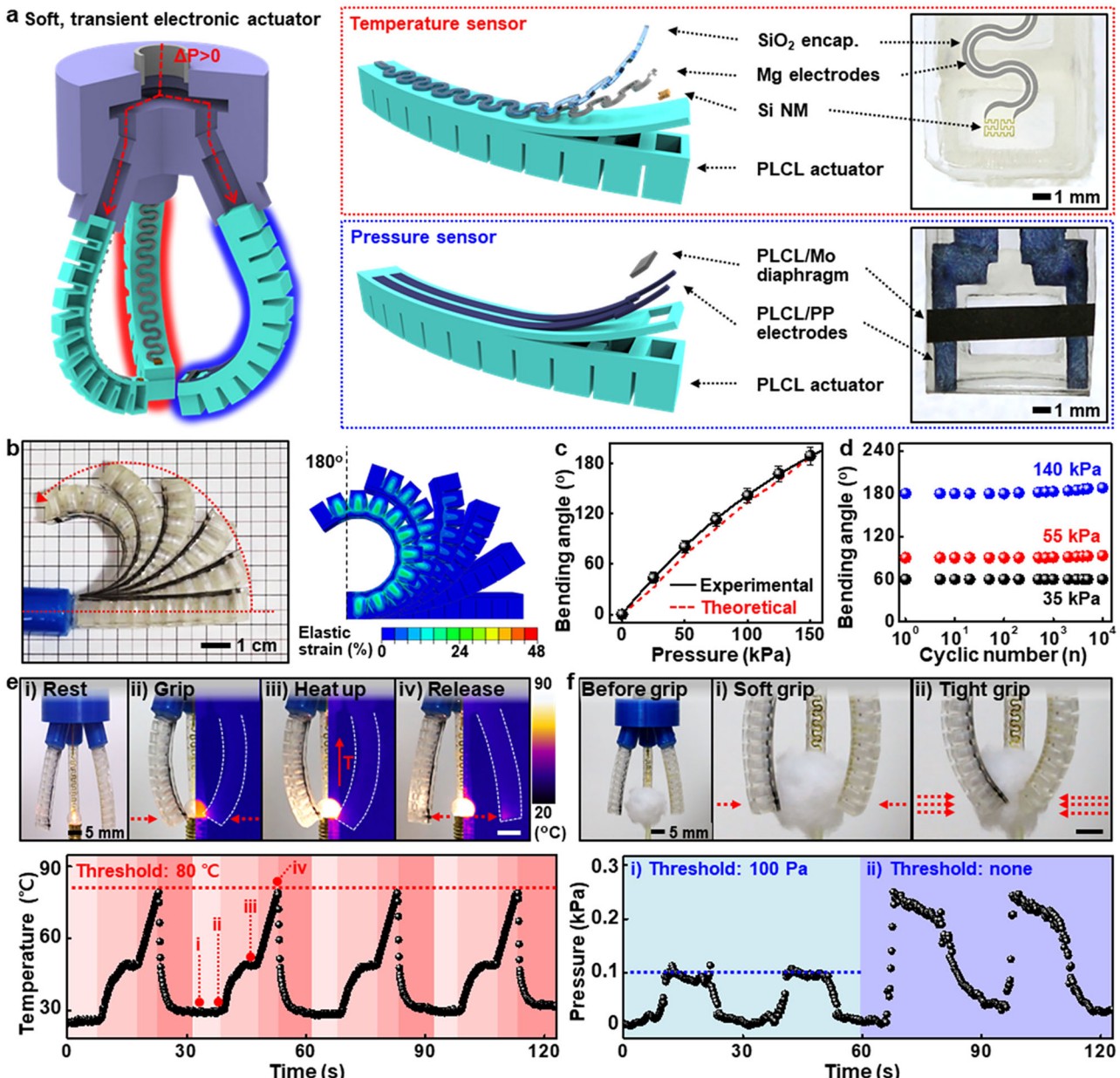

**Fig. 4 | Smart transient soft robots with integrated electronics. a** Description of a PLCL-based transient robotic gripper capable of perceiving environmental factors such as temperature and pressure via integrated electronic components (left) and schematic exploded (middle) and magnified (right) views of individual sensing elements. **b** Photograph (left) and finite element simulation result (right) at several states of bending actuation of the gripper. When bent to 180°, the gripper experienced only a strain of less than 20%, corresponding to the elastic region of PLCL. **c** Relationship between bending angle and applied pneumatic pressure. Data are presented as mean values ± standard deviation. $n = 3$ independent samples. **d** Cyclic actuation stability of the gripper under several pressures of 35, 55, and 140 kPa. **e** Overall procedures of temperature-aware actuation with a threshold value of 80 °C (top: (i) rest, (ii) grip, (iii) heat up, and (iv) release), and controlled temperature condition of the bulb over time (bottom). **f** Comparison of the behavior of pressure sensor-embedded grippers with a limited pressure of 100 Pa for gripping soft, light cotton balls (top) and the corresponding changes of pressure values ((i) -100 Pa, (ii) none).

performances of this transient system can address drawbacks of short operation lifetimes or non-degradability reported in previous articles[4,19,61]. As an indicator to observe cardiac dynamics during the early stages of heart diseases (e.g., myocardial infarction), PLCL/Mo composites could provide information on heart rate and can be further utilized to predict myocardial strain/stiffness when integrated with an appropriate encapsulation strategy (Supplementary Fig. 31). The effect of temperature on the sensitivity of PLCL/Mo composite appears in Supplementary Note 11 and Fig. 32. We could not observe substantial physical degradation of the device throughout the implantation period of 8 weeks due to the slow dissolution rate. However, as shown in the decrease in sensitivity of PLCL/Mo composite after 2 weeks of

implantation in Supplementary Fig. 31, Mo flakes were being degraded by water molecules penetrating the encapsulation layer at a rate of ~20 nm/day[62], without remaining harmful byproducts. Other constituent materials, such as PEDOT:PSS and P14[TFSI], would be gradually disintegrated as the PLCL matrix dissolved, while the long-term biocompatibility of these materials requires further validation.

To evaluate the cytotoxicity of PLCL-based cardiac-integrated electronics and byproducts upon dissolution, we assessed histological examination of myocardial tissues by hematoxylin and eosin (H&E) staining (Fig. 5f and Supplementary Fig. 33). During 8 weeks of implantation, fibrosis progressed on the epicardial outer surface at implantation sites while there was no observable evidence of

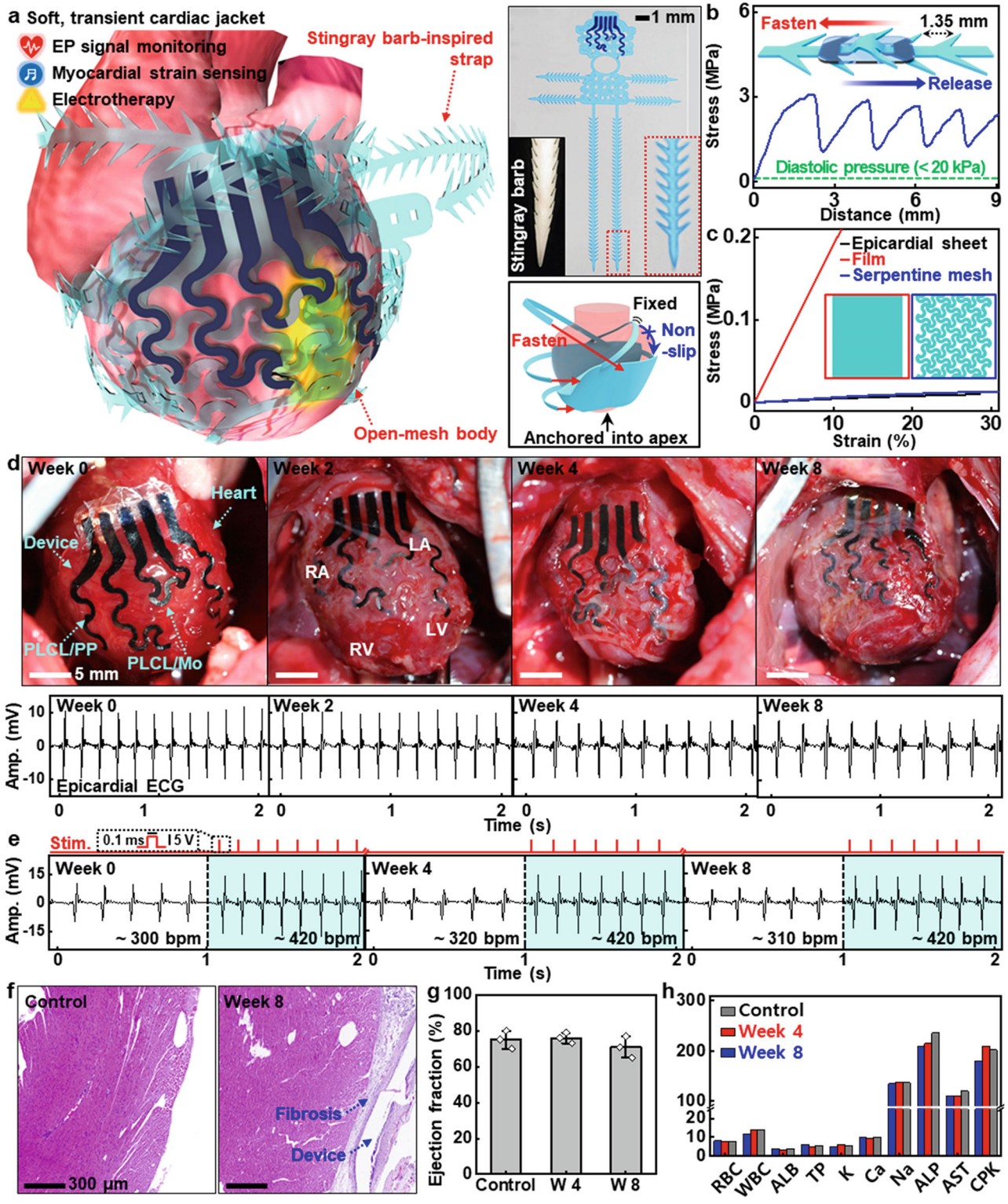

damage in cardiac tissues. Hemodynamic studies revealed no significant responses in left ventricular ejection fractions (LVEFs), indicating that the implanted electronic systems induced no adverse effects on cardiac functions (Fig. 5g). In complete blood count and blood chemistry analysis, all the levels fall within confidence intervals of control values throughout 8 weeks of implantation, confirming the absence of disorders in organs and overall health conditions (Fig. 5h, additional results and postoperative weight changes of animals appear in Supplementary Fig. 34). Together with previous reports on non-toxicity of constituent materials[33,50,51,63], these comprehensive assays provide strong evidence of the biocompatibility of the device.

## Discussion

Synthesis, comprehensive characterizations, and a set of applications presented here illustrate the outstanding mechanical/physical properties of the biodegradable, stretchable PLCL elastomers for eco-friendly and biologically safe electronic devices. Physical combinations with organic/inorganic conductive fillers yielded conductive polymer composites to enable the rich versatility of electronic components.

**Fig. 5 | Implantable, multifunctional, transient suture-free cardiac jacket for heart diseases. a** Schematic illustration of a soft, transient cardiac jacket consisting of PLCL-based open-mesh body, stingray barb-inspired straps, and sensing/therapeutic components (PLCL/PP for both epicardial ECG monitoring and electrical stimulation on left ventricle; PLCL/Mo for myocardial strain sensing), with photographs of a cardiac jacket fabricated with blue dye for clear visualization and a stingray barb (top right) and a description of suture-free integration of the device on the heart (bottom right). **b** Releasing force of the strap for facile and robust integration of the device to the heart, where the maximum releasing force was ~3 MPa at a displacement rate of 1 mm/s. **c** Mechanical modulus matching of the cardiac jacket with a rat heart via the use of a mesh structure to achieve conformal contact and avoid heart failures. **d** Photographs of the implanted cardiac jackets (top) and representative epicardial ECG signals recorded by the devices (bottom) on weeks 0, 2, 4, and 8 of in vivo implantation. **e** Epicardial ECG signals measured before (left white background) and during (right blue background) electrical

stimulation (square wave with a duration of 0.1 ms and an amplitude of 5 V) with the implanted cardiac jackets on weeks 0, 4, and 8 of implantation. **f** Representative hematoxylin and eosin (H&E) stained cross-sectional images of the left ventricle of a rat without (left) and with device implantation for 8 weeks (right). **g** Negligible changes in ejection fraction in rats before (control) and 4 and 8 weeks after device implantation, which show that the device did not adversely affect cardiac functions. $n = 3$ biologically independent animals per group. Data are presented as mean values ± standard deviation. **h** Analysis of complete blood counts and blood chemistry for rats with and without device implantation, indicating preservation of overall healthy physiology in the animals. $n = 3$ biologically independent animals per group. RBC red blood cell count, K/uL; WBC white blood cell count, K/uL; ALB albumin, g/dL; TP total protein, g/dL; K potassium, mmol/L; Ca Calcium, mmol/L; Na sodium, mmol/L; ALP alkaline phosphatase, U/L; AST aspartate aminotransferase, U/L; CPK creatine phosphokinase, U/L.

Different structural designs and integration with diverse functional units could create advanced forms of soft robots that can conduct more intricate/sophisticated tasks in vivo or under harsh environmental conditions, and a non-invasive medical implant that can accomplish sensing/therapeutic functions for various cardiac diseases, such as arrhythmia, tachycardia, fibrillation, and myocardial infarction, over long periods of time. The proposed materials and elaborate engineering strategies are expected to appear further in environmental monitoring systems, edible devices, and bio-integrated electronics, paving the way for broadening the field of transient electronics.

## Methods

All animal experiments were approved by the Institutional Animal Care and Use Committee (approval number: KOREA-2022-0022), Korea University, and followed the ethical principles for animal experimentation established by the institute.

### Synthesis of poly(L-lactide-co-ε-caprolactone) (PLCL)

Predetermined amounts of ε-caprolactone (ε-CL, Alfa Aesar, USA; 500 mmol), L-lactide (LA, Medichem, South Korea; 500 mmol), stannous octoate (Sn(Oct)$_2$, Alfa Aesar, USA; 5 mmol), and 1-dodecanol (Alfa Aesar, USA; 0.2, 0.1, and 0.05 mmol for molecular weights of 60k, 140k, and 200k, respectively) were added into a dried glass ampule equipped with a magnetic stirring bar. The ampule was sealed under vacuum after purging three times with $N_2$ at 90 °C and heated in an oil bath at 150 °C. After the reaction for 24 h, the product was dissolved in chloroform and microfiltered through a membrane filter with a pore size of 0.45 μm. It was then precipitated by pouring the polymer solution into excess methanol to remove the catalyst and unreacted monomers, filtered, and dried in a vacuum oven at 50 °C for 24 h. The products were analyzed by $^1$H nuclear magnetic resonance (NMR) and gel permeation chromatography (GPC) (Supplementary Fig. S1).

### PLCL/PEDOT:PSS-based partially biodegradable, conductive polymer composites (PLCL/PP)

Poly(3,4-ethylenedioxythiophene):poly(styrenesulfonate) (PEDOT:PSS, Clevios PH1000, Heraeus GmbH, Germany) was dried in a freeze-dryer (FD 8518, ilShinBioBase, South Korea), added with dimethylsulfoxide (DMSO, Sigma-Aldrich) at a concentration of 1 w/v%, and homogeneously dispersed by a probe sonicator (VCX-500, Sonics & Materials, USA) for 3 h with a 10-s on/off interval. Appropriate volumes of 1 w/v% PLCL (200k) solution dissolved in DMSO were mixed with the PEDOT:PSS dispersion to make PLCL/PP ink at weight ratios of PLCL and PEDOT:PSS varying from 0.8:0.2 to 0:1.0, with 1 weight ratio of an ionic liquid, N-methyl-N-butylpyrrolidinium bis(trifluoromethanesulfonyl) imide (P14[TFSI], Sigma-Aldrich, USA). For studying the effect of P14[TFSI] on the conductivity and stretchability of PLCL/PP, different weight ratios (0, 0.5, 1, 2, and 3)

of P14[TFSI] were added into the PLCL/PP ink with a weight ratio of 0.6:0.4. All inks were magnetically stirred at RT for 12 h, drop-casted into an oxygen plasma-treated PDMS mold, and dried at 80 °C for 12 h. For a supporting layer, a 15 w/v% PLCL (200k) solution dissolved in DMF was further drop-casted into the mold, dried at 80 °C for another 12 h, and mechanically removed from the mold for characterizations. Conductivity and stretchability measurements were carried out on three different samples for each weight ratio, using a home-built tensile tester connected with a four-point probe system (M4P205, MStech, South Korea) at a stretching rate of 6 mm/min. Conductivity was calculated by the equation: electrical conductivity (S/cm) = 1 / (sheet resistance (Ω/sq) × thickness (cm)), where the sheet resistance and the thickness were defined by the four-probe method and SEM analysis. A degradation test of PLCL/PP (0.6:0.4:1) was conducted in PBS (pH 7) at 37 °C while monitoring the change in conductivity.

### PLCL/Mo flake-based strain-sensitive conductive composites (PLCL/Mo)

Fabrication of PLCL/Mo began with the preparation of Mo flakes. A Mo film (500 nm thick) was sputtered onto a PMMA layer (~100 nm thick) spin-coated on a Si wafer and released from the wafer by immersing in acetone, followed by applying ultrasonication to break the film into flakes with an average diameter of ~10 μm, which was characterized by optical microscopy. The Mo flakes were then sedimented in a 50-mL tube and rinsed with a fresh DMF three times to remove a residual PMMA, followed by mixing with appropriate volumes of the 5 w/v% PLCL (200k) solution to make PLCL/Mo inks at volume ratios of PLCL and Mo flake varying from 9:1 to 7:3. The PLCL/Mo inks were drop-casted into prefabricated PDMS molds, dried at 80 °C for 12 h, and peeled off from the molds for characterizations and applications. Morphologies and microstructures were analyzed by SEM, and electrical properties were characterized by measuring three different samples for each volume ratio, using a home-built tensile tester connected with a four-point probe system at a stretching rate of 6 mm/min. A degradation test of PLCL/Mo (9:1) was conducted in PBS (pH 7) at 37 °C while monitoring the change in sheet resistance.

### Transient electronic actuator

Fabrication of a transient, smart soft actuator involved three steps: fabricating temperature/pressure sensors on a PLCL layer, forming a semi-open PLCL actuator, and their assembly (detailed structures in Supplementary Fig. S19). Similar to the fabrication processes for the CMOS array, a lightly doped Si NM with two highly doped regions for electrode contacts was formed and transfer-printed onto PMMA/PI bilayers coated on a Si wafer, followed by patterning the Si NM by RIE and the subsequent deposition/patterning layers of Mg for electrodes and $SiO_2$ for encapsulation. Removing the PMMA layer in acetone enabled the device to be picked up by a PDMS stamp. Dry etching the

bottom PI layer, transfer-printing the device onto a PLCL layer (140k, ~500 μm-thick), and dry etching the top PI layer completed the temperature sensor. Attaching a free-standing piece of PLCL/Mo film (9:1; ~100 μm-thick) onto the hole of a PLCL layer (140k, ~500 μm-thick) integrated with PLCL/PP electrodes (0.6:0.4:1) completed the diaphragm-based pressure sensor. Finally, ACF cables were connected to the electrodes of the sensors to monitor changes in resistance by the source meter. The semi-open actuator with 10 pneumatic chambers was molded by repeatedly drop-casting a 15 w/v% of PLCL solution (140k) dissolved in DMF into a PDMS mold and drying at 80 °C. The pneumatic chambers were divided by sharp cutting the wall between them with a razor blade. The semi-open actuator was bonded to the PLCL layer integrated with temperature or pressure sensors by applying a small amount of the PLCL solution and then assembled with a 3D-printed PLA head connecting to a pneumatic source. For a soft gripper based on high and low moduli of PLCLs, 60k and 200k PLCLs were used to fabricate a semi-open actuator and a top layer of the gripper, respectively.

## Soft, transient, suture-free cardiac-integrated electronics

The prepared PLCL/PP ink (PLCL:PEDOT:PSS:P14[TFSI], 0.6:0.4:1) and PLCL/Mo ink (PLCL:Mo flake, 9:1) were screen-printed on a PDMS-casted acrylic plate through laser-cut PDMS stencil masks (200 μm-thick), followed by drying on a hotplate at 80 °C for 12 h. After removal of the masks, the resultant PLCL/Mo (~30 μm-thick) was transfer-printed on the PLCL/PP electrodes (~5 μm-thick) by applying 1 μL of the PLCL/PP ink to form a strain sensor. Then, 20 w/v% PLCL solution (140k) was poured and dried at 80 °C overnight. Laser-cutting defined layouts of a mesh device (~50 μm-thick), including the PLCL/PP and the PLCL/Mo and an encapsulation layer with openings for EP sensing, electrical stimulation, and wiring. The mesh device and encapsulation layer were aligned and assembled by applying heat (80 °C) and pressure (~100 kPa) for 3 h. Bonding of one side of Mo wires (diameter, 100 μm; Shanghai QUKEN New Material Technology, China) to contact pads of the PLCL/PP electrodes using viscous PLCL/PP ink (30 wt%) and PLCL solution (50 w/v%) and the other side to a pin connector (PH03-D80P, Eleparts, South Korea) by soldering and encapsulation with Ecoflex completed the fabrication.

## Reporting summary

Further information on research design is available in the Nature Portfolio Reporting Summary linked to this article.

## Data availability

All data supporting the findings and conclusions of this study are available within the paper and its Supplementary Information files. All other relevant data are available from the corresponding author upon request.

## Code availability

A custom LabVIEW program for the measurements of EMG and surface and epicardial ECGs and a custom Aduino Genuino program for the operation of 8×16 RGB LED matrix array are freely available for download at http://github.com/gwanjinko.

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

## Acknowledgements

This work was supported by a Korea University grant, the KU-KIST Graduate School of Converging Science and Technology Program, the National Research Foundation of Korea (NRF) grant funded by the Kor-ean government (the Ministry of Science, ICT, MSIT) (RS-2022-00165524) (to S.-W.H.), the Korea Medical Device Development Fund Grant funded by the Korea government (the Ministry of Science and ICT, the Ministry of Trade, Industry and Energy, the Ministry of Health & Welfare, and the Ministry of Food and Drug Safety) (1711174536, RS-2020-KD000138) (to S.-W.H.), and the Ministry of Science and ICT(MSIT), under the ICT Creative Consilience program (IITP-2023-2020-0-01819) supervised by the IITP (Institute for Information & Communications Technology Planning & Evaluation) (to S.-W.H.).

## Author contributions

W.B.H., G.-J.K., and S.-W.H. conceived and designed the research and interpreted the results. W.B.H. and G.-J.K. led the experiments. W.B.H., G.-J.K., and K.-G.L. performed in vivo surgery and associated pre- and postoperative procedures with support from Y.-D.P. D.K., J.H.L., S.M.Y., D.-J.K., J.-W.S., T.-M.J., S.H., H.K., J.H.L., and K.R. contributed to the fabrication and characterization of materials and devices. H.Z. con-ducted the theoretical analysis with support from H.C. S.H.K. supported the experiments and commented on the paper. W.B.H., G.-J.K., and S.-W.H. co-wrote the manuscript. All authors read and approved the final manuscript.

## Competing interests

The authors declare no competing interests.
