## [Peer review file · Nature Communications]

REVIEWER COMMENTS

Reviewer #1 (Remarks to the Author):

The authors investigated biodegradable elastomer PLCL and explored associated applications as soft grippers and cardiac jackets. The work is interesting and can be considered as publication in Nat. Commun. after addressing some issues:

(1) The authors need to elucidate the novelty of the materials presented in the work, as PLCL has already been reported by several papers before. The authors also need to elaborate why it is advantageous to use PLCL for biodegradable electronics over other stretchable and biodegradable polymers or hydrogels, as there are already many reported demonstrations based on these stretchable materials.

(2) The introduction is too brief on materials discussions. Although the authors claimed that existing biodegradable and stretchable polymers “are far from practical and multipurpose uses due to insufficient characteristics, such as uncontrollable, dissolution behaviors, limited mechanical elasticity/modulus/vulnerability, or poor storage stability”, no evidence is given which makes the statement quite unconvincing.

(3) Fig 2i, how about the comparison with biodegradable and stretchable hydrogels? For example, such as the following reported work, ref: Strong tough hydrogels via the synergy of freeze-casting and salting out. Nature 590, 594–599 (2021). <https://doi.org/10.1038/s41586-021-03212-z>; Photocurable bioresorbable adhesives as functional interfaces between flexible bioelectronic devices and soft biological tissues. Nat. Mater. 20, 1559–1570 (2021). <https://doi.org/10.1038/s41563-021-01051-x>.

(4) To achieve biodegradable conductors, the authors mixed PEDOT:PSS and ionic liquid with PLCL. PEDOT:PSS is not dissolvable in aqueous solutions, and ionic liquids are often cytotoxic. The authors need to provide detailed evidence and discussions, otherwise bioresorbable electronics based on these materials are not feasible.

(5) Methodology regarding animal experiments for cardiac jacket are missing. For example, how were the devices deployed to the hearts of rats? How were the devices connected to external equipment for measurement over the implantation period? How was the connection protected from damage by the rats as biting or scratching of wire connectors often occur? Is the current device connection strategy feasible for clinical settings?

(6) Fig. 5a, strain sensors are involved in the device. Temperature changes indicating heart malfunctions can also affect the output signals of the sensor. How to differentiate these two parameters?

(7) The authors demonstrated ECG recording and electrical stimulation using the cardiac jacket. But it is not clear what clinical diagnostic functions or treatment the device can deliver. For example, the authors need to elaborate how can the device predict myocardial stiffness as claimed in the paper.

(8) Fig. 3g, the authors stated that “the stretchability and conductivity far surpass results in previous reports”, which seems not true as the reported stretchability of ref 46 looks better than the current work.

(9) Other issues, in the methodology part, Mo films ~ 500 micrometers were sputtered onto PMMA. Is sputtering such thick Mo films a scalable process?

Reviewer #2 (Remarks to the Author):

In this manuscript, a highly stretchable and biodegradable elastomer is synthesized as the substrate for soft transient electronics. Biodegradable transient devices show promising potentials in implantable bioelectronics and environmental-friendly electronics. A series of eye-catching demonstrations are presented in the manuscript to illustrate the potential applications of the elastomer. Despite the interesting aspects, I find several major inconsistencies in the study involving non-degradable/partially degradable devices. If the new property is not harnessed, the cool demonstrations may not be very relevant. Tremendous work is likely required to fix many scientific/technical issues. Otherwise, the study may not qualify for this prestigious journal.

Major scientific inconsistency:

1. An LED array is the first device demonstration. Most electronic components are non-degradable materials, including PMMA, PI, Surface mount chips, silver epoxy, etc. I am surprised that the demonstration is standard electronics on a biodegradable substrate. I cannot find any major benefits of this device system for practical applications.
2. A conductive composite is prepared as a mixture of PEDOT:PSS, P14[TFSI], and PLCL. PEDOT:PSS is a well-recognized biocompatible polymer rather than a biodegradable one. The degradability of the plasticizer is also open to question. With limited biodegradability, the conductive composite is better considered a transient conductor.
3. The conductive composite is used to construct an electronic mesh for cardiac monitoring and stimulation. Additional experiments deal with biocompatibility. Similar results are easily obtained with standard silicone (e.g., Ecoflex). The in vivo experiments have nothing to do with the biodegradable/transient characteristics of the elastomer.

4. The biodegradable elastomer has not been effectively characterized with benign reaction products suitable for implantable applications. The experimental tests actually deal with environmental degradation.

Technical issues,

1. Poly(L-lactide-co- ϵ -caprolactone) (PLCL) is widely used in tissue engineering with limited stretchability. The underlying mechanism of the improved stretchability observed here should be further clarified. In design and synthesis, the differences from previous reports should also be explicitly explained.

2. The reaction mechanism of the degradation process should be explicitly explained to justify the products' benign nature.

3. It is better to explain the conductivity calculation of stretched conductive composites. The claim on strain-insensitive characteristics lacks any experimental evidence. In Fig 3d, the emission intensity drops in response to large tensile deformations. The device is obviously very sensitive to strains.

4. A PLCL/Mo composite is employed for strain sensing. Within a minor strain of 2%, the resistance increases by one order of magnitude over several hundred cycles. The significant drift contradicts the claim of reliable cyclic behaviors (Line 175). In addition, the limited cycling stability is a practical issue for cardiac strain monitoring involving excessive cycles. In Fig S30, the responses show obvious deteriorations over time.

Reviewer #3 (Remarks to the Author):

The authors use poly(l-lactide-co- ϵ -caprolactone) (PLCL) as an elastomer for transient electronics. They demonstrate in detail a variety of device work and some degradation studies. Figures are well made and the experiments are thorough. I see the major advance of this paper is fabricating electronics on PLCL, which has not been done before. However, I do not consider PLCL a new elastomer system. That said, I believe this work merits publication in nature communications once the comments below are addressed.

In the introduction, the authors mention that “recent research efforts have introduced a few different types of biodegradable and stretchable materials based on protein, gelatin, and

crosslinked polymers". I recognize the authors attempt to highlight that there are limited polymers used for transient electronics, but it gives the impression that there is little work in this area for non-experts not familiar with the polymer field. There is rich literature on biodegradable elastomers, and detailed analysis of degradation. PLCL has been heavily studied, and there are studies of its use as substrates for electronic materials—people have blended it with inorganic/organic materials and even co-polymerized conductive polymers on its substrate. Often, researchers outside of the electronics-focused or applications-focused community omit these perspectives; I think it would be powerful to acknowledge these works.

In figure 2, the authors highlight the improvement of PLCL elastomers compared to other works. There likely are other biodegradable elastomers with similar modulus (2e) that can also be stretched to 1000%. This graph also shows hysteresis at 40%, which is exaggerated at higher strains (SI Fig 5), and plastic deformation-- thank you for including this additional characterization.

In figure 3g, the authors compare their work with other reported literature. Notably, their work has comparable conductivity to other literature but can stretch to ~600% instead of 300% (references 41 and 42). While this is impressive, it is unclear what the application for this is, and whether this increase is necessary. This is highlighted by the types of device demonstrations that do not show why 600% is better than 300%. Some comments on this point in the manuscript would be appreciated.

Reviewer #1 (Remarks to the Author):

The authors investigated biodegradable elastomer PLCL and explored associated applications as soft grippers and cardiac jackets. The work is interesting and can be considered as publication in Nat. Commun. after addressing some issues:

Our response: We thank the reviewer for the positive assessment and for the recommendation to publish in *Nature Communications*. We believe that the improved manuscript, through feedback on the reviewer's valuable comments, will inspire the journal community and readers.

Comment #1: The authors need to elucidate the novelty of the materials presented in the work, as PLCL has already been reported by several papers before. The authors also need to elaborate why it is advantageous to use PLCL for biodegradable electronics over other stretchable and biodegradable polymers or hydrogels, as there are already many reported demonstrations based on these stretchable materials.

Our response: In the material aspect, most previous works on PLCL exhibit limited stretchability (< 500 %) and high mechanical modulus (> 10 MPa) primarily due to high L-lactide (LA) to ϵ -caprolactone (CA) ratio, which is not quite appropriate for integration of soft tissues and organs with electronic components. In this work, we synthesized PLCL elastomers with a 5:5 ratio to improve stretchability and decrease mechanical modulus, and diversified the mechanical/physical/biochemical properties via varying the ratio of monomers to initiator.

We introduced various demonstrations based on an unexplored geometry, e.g., thin film, and characterized the relevant mechanical/physical/biochemical properties for integration of soft, ultrathin electronic components, including elasticity, toughness, fracture tolerance, transparency, adhesion, and chemical resistance, and we highlighted promising applicability of PLCL elastomers for versatile transient systems.

Although PLCL has been widely used in the form of a porous membrane as a biological scaffold, such porous structures might not be suitable for integration with complex, sophisticated electronic devices and systems.

It is difficult for existing biodegradable polymers to simultaneously satisfy the various properties listed above, and in general, hydrogels have issues of operation in a dry environment due to their moisture-, water-rich feature. For comparison, we have provided a supplementary table including many examples of biodegradable polymers and hydrogels with their mechanical/biochemical properties.

Our modifications: We have added the following table and related references to the Supplementary Information.

Materials	T _g [°C]	T _m [°C]	Young's modulus [E, MPa]	Elongation at break [ε _b , %]	Degradation rate [weight loss in time]	Shelf life [in ambient conditions]	Issues	Ref	
PLCL (50/50)	-7 - -3	-	2.5 - 7.5	900 - 1600	20 % in 30 weeks	> 6 month		Our works	
Silk	-	-	10000	20	35 % in 3 weeks	-	High modulus Poor stretchability	[1]	
PLA	55	183	3000 - 3500	5 - 30	several years	5 years	High modulus Poor stretchability	[2,3]	
PCL	-60	60-62	297 - 363	700-1000	several years	5 years	High modulus Poor elasticity	[2,4,5]	
Biodegradable polymers	PGA	35-45	220	7000 - 8400	30	6 - 12 month	5 years	High modulus Poor stretchability	[4]
	PLGA (50/50)	40-60	153	2000	3 - 10	1 - 2 weeks	4 weeks in dry condition	High modulus Poor stretchability Rapid degradation	[3,6]
	PBS	5	163	700	6	10 % in 14 weeks	-	High modulus Poor stretchability	[7,8]
	PHB	5	180	3500	5-8	60 % in 24 weeks	-	High modulus Poor stretchability	[9]
	POC	0.2	-	0.92 - 16.4	260	50 % in 25 weeks	-	Low stretchability	[10]
	b-DCPU	-35 - -30	-	0.5 - 3.8	230	20 % in 10 weeks	-	Low stretchability	[11]
	PGS	-51.24, -18.50	5.23, 37.62	0.5	330	17 % in 60 days	-	Low stretchability	[12]
	PSeD-U	-20 - -2	-	0.54 - 0.74	136 - 509	60% in 8 hrs	-	Rapid degradation	[13]
	PGSA	-	-	0.143 - 0.592	20 - 130	90 % in 8 weeks	12 months	Low stretchability	[14]
	Gelatin biogel	-	-	0.03 - 0.3	320	60 % in 4 days	13 months	Low stretchability Rapid degradation	[15]
Hydrogels	TRn11	-	-	2.4	300	on demand	-	Low stretchability Poor storage stability	[16]
	GelMa	-	-	0.01 – few tens	150 - 290	10 days	6 months	Low stretchability Rapid degradation Poor storage stability	[17,18]
	HA-PVA	-	200-250	2.5	2900	N/A	-	Poor degradation Poor storage stability	[19]
	BTIM	-	-	0.03	1200	20 days	-	Rapid degradation Poor storage stability	[20]

Table S1. Summary of representative biodegradable polymers and hydrogels with their mechanical/degradation properties

[1] Vepari, C. & Kaplan, D. L. Silk as a biomaterial. *Prog. Polym. Sci.* **32**, 991–1007 (2007).

[2] Sarazin, P., Roy, X. & Favis, B. D. Controlled preparation and properties of porous poly (L-lactide) obtained from a co-continuous blend of two biodegradable polymers. *Biomaterials* **25**, 5965-5978 (2004).

[3] Gentile, P. et al. An Overview of Poly(lactic-co-glycolic) Acid (PLGA)-Based Biomaterials for Bone Tissue Engineering. *Int. J. Mol. Sci.* **15**, 3640-3659 (2014).

[4] Manavitehrani, I. et al. Biomedical Applications of Biodegradable Polyesters. *Polymers* **8**, 20 (2016).

[5] Eshraghi, S. & Das, S. Mechanical and microstructural properties of polycaprolactone scaffolds with one-dimensional, two-dimensional, and three-dimensional orthogonally oriented porous architectures produced by selective laser sintering. *Acta Biomater.* **6**, 2467-2476 (2010).

[6] Houchin, M. L., & E. M. Topp. Physical Properties of PLGA Films During Polymer Degradation. *J. Appl. Polym. Sci.* **114**, 2848-2854 (2009).

[7] Lindström, A., Albertsson, AC. & Hakkarainen, M. Quantitative determination of degradation products an effective means to study early stages of degradation in linear and branched poly (butylene adipate) and poly (butylene succinate). *Polym. Degrad. Stab.* **83**, 487-493 (2004).

- [8] Liu, L. et al. Mechanical properties of poly (butylene succinate) (PBS) biocomposites reinforced with surface modified jute fibre. *Compos. - A: Appl. Sci. Manuf.* **40**, 669-674 (2009).
- [9] Volova, T. et al. Results of biomedical investigations of PHB and PHB/PHV fibers. *Biochem. Eng. J.* **16**, 125-133 (2003).
- [10] Yang, J., Webb, A. R. & Ameer, G. A. A. Novel citric acid-based biodegradable elastomers for tissue engineering. *Adv. Mater.* **16**, 511-516 (2004).
- [11] Choi, Y. S. et al. Stretchable, dynamic covalent polymers for soft, long-lived bioresorbable electronic stimulators designed to facilitate neuromuscular regeneration. *Nat. Commun.* **11**, 1–14 (2020).
- [12] Wang, Y. et al. A tough biodegradable elastomer. *Nat. Biotechnol.* **20**, 602-606 (2002).
- [13] Chen, S. et al. Mechanically and biologically skin-like elastomers for bio-integrated electronics. *Nat. Commun.* **11**, 1–8 (2020).
- [14] Held, M. et al. Soft electronic platforms combining elastomeric stretchability and biodegradability. *Adv. Sustain. Syst.* **6**, 2100035 (2022).
- [15] Baumgartner, M. et al. Resilient yet entirely degradable gelatin-based biogels for soft robots and electronics. *Nat. Mater.* **19**, 1102–1109 (2020).
- [16] Pena-Francesch, A., Jung, H., Demirel, M. C. & Sitti, M. Biosynthetic self-healing materials for soft machines. *Nat. Mater.* **19**, 1230–1235 (2020).
- [17] Van Den Bulcke, A. I. et al. Structural and rheological properties of methacrylamide modified gelatin hydrogels. *Biomacromolecules* **1**, 31-38 (2000).
- [18] Kong, B. et al. Fiber reinforced GelMA hydrogel to induce the regeneration of corneal stroma. *Nat. Commun.* **11**, 1-12 (2020).
- [19] Hua, M. et al. Strong tough hydrogels via the synergy of freeze-casting and salting out. *Nature* **590**, 594-599 (2021).
- [20] Yang, Q. et al. Photocurable bioresorbable adhesives as functional interfaces between flexible bioelectronic devices and soft biological tissues. *Nat. Mater.* **20**, 1559-1570 (2021).

Comment #2: The introduction is too brief on materials discussions. Although the authors claimed that existing biodegradable and stretchable polymers “are far from practical and multipurpose uses due to insufficient characteristics, such as uncontrollable, dissolution behaviors, limited mechanical elasticity/modulus/vulnerability, or poor storage stability”, no evidence is given which makes the statement quite unconvincing.

Our response: We thank the reviewer for this comment. As described in the manuscript, biodegradable polymers with excellent mechanical/biochemical properties are essential for further developments in extended, sophisticated functionality or systems for bioresorbable, transient applications. However, natural biodegradable polymer, silk [1], and commercially available biodegradable polymers, such as poly(l-lactide) (PLA) [2,3], poly(glycolide) (PGA) [4], poly(lactide-co-glycolide) (PLGA) [3,5], poly(butylene succinate) (PBS) [6,7], and poly(3-hydroxybutyrate) (PHB) [8] had very high modulus (several hundreds to thousands MPa) and inelasticity with maximum elongation of 3 ~ 30 %. Although poly(ϵ -

caprolactone) (PCL) showed an elongation at break of 700 ~ 1000 %, the material still exhibited high modulus (297 ~ 363 MPa) and irreversible plasticity [9].

Biodegradable elastomers, such as poly(1,8-octanediol-co-citrate) (POC) [10], bioresorbable dynamic covalent polyurethane (b-DCPU) [11], and poly(glycerol-sebacate) (PGS) [12] had relatively lower modulus of 0.5 ~ 16.4 MPa, however their maximum stretchability was in the range of 230 ~ 330 %. Poly(sebacoyl diglyceride)-graft-2-ureido-4[1H]-pyrimidinone unit (PSeD-U) showed skin-like modulus of 0.54 ~ 0.74 MPa and maximum stretchability of ~500 %, while a rapid dissolution rate (60 % degraded in 8 h under PBS at 37 °C) with a narrow control margin restricted potential research areas [13]. Biodegradable hydrogels based on various materials such as poly(glycerol sebacic)acrylate (PGSA) [14], gelatin biogel [15], biosynthetic proteins with 11 tandem repetitions of the squid-inspired building block (TRn11) [16], gelatin methacrylate (GelMA) [17,18], hierarchical and anisotropic structures poly(vinyl alcohol) (HA-PVA) [19], and bioelectronic-tissue interface material (BTIM) [20] in recent studies exhibited low Young's modulus and enhanced stretchability. However, the nature of hydrogels -- which inevitably contains water -- led to not only difficulty in integrating with electronic devices but also poor storage stability. All these issues have been summarized in the supplementary table above.

- [1] Vepari, C. & Kaplan, D. L. Silk as a biomaterial. *Prog. Polym. Sci.* **32**, 991–1007 (2007).
- [2] Sarazin, P., Roy, X. & Favis, B. D. Controlled preparation and properties of porous poly (L-lactide) obtained from a co-continuous blend of two biodegradable polymers. *Biomaterials* **25**, 5965-5978 (2004).
- [3] Gentile, P. et al. An Overview of Poly(lactic-co-glycolic) Acid (PLGA)-Based Biomaterials for Bone Tissue Engineering. *Int. J. Mol. Sci.* **15**, 3640-3659 (2014).
- [4] Manavitehrani, I. et al. Biomedical Applications of Biodegradable Polyesters. *Polymers* **8**, 20 (2016).
- [5] Houchin, M. L., & E. M. Topp. Physical Properties of PLGA Films During Polymer Degradation. *J. Appl. Polym. Sci.* **114**, 2848-2854 (2009).
- [6] Lindström, A., Albertsson, AC. & Hakkarainen, M. Quantitative determination of degradation products an effective means to study early stages of degradation in linear and branched poly (butylene adipate) and poly (butylene succinate). *Polym. Degrad. Stab.* **83**, 487-493 (2004).
- [7] Liu, L. et al. Mechanical properties of poly (butylene succinate) (PBS) biocomposites reinforced with surface modified jute fibre. *Compos. - A: Appl. Sci. Manuf.* **40**, 669-674 (2009).
- [8] Volova, T. et al. Results of biomedical investigations of PHB and PHB/PHV fibers. *Biochem. Eng. J.* **16**, 125-133 (2003).
- [9] Eshraghi, S. & Das, S. Mechanical and microstructural properties of polycaprolactone scaffolds with one-dimensional, two-dimensional, and three-dimensional orthogonally oriented porous architectures produced by selective laser sintering. *Acta Biomater.* **6**, 2467-2476 (2010).
- [10] Yang, J., Webb, A. R. & Ameer, G. A. Novel citric acid-based biodegradable elastomers for tissue engineering. *Adv. Mater.* **16**, 511-516 (2004).
- [11] Choi, Y. S. et al. Stretchable, dynamic covalent polymers for soft, long-lived bioresorbable electronic stimulators designed to facilitate neuromuscular regeneration. *Nat. Commun.* **11**, 1–14 (2020).

- [12] Wang, Y. et al. A tough biodegradable elastomer. *Nat. Biotechnol.* **20**, 602-606 (2002).
- [13] Chen, S. et al. Mechanically and biologically skin-like elastomers for bio-integrated electronics. *Nat. Commun.* **11**, 1–8 (2020).
- [14] Held, M. et al. Soft electronic platforms combining elastomeric stretchability and biodegradability. *Adv. Sustain. Syst.* **6**, 2100035 (2022).
- [15] Baumgartner, M. et al. Resilient yet entirely degradable gelatin-based biogels for soft robots and electronics. *Nat. Mater.* **19**, 1102–1109 (2020).
- [16] Pena-Francesch, A., Jung, H., Demirel, M. C. & Sitti, M. Biosynthetic self-healing materials for soft machines. *Nat. Mater.* **19**, 1230–1235 (2020).
- [17] Van Den Bulcke, A. I. et al. Structural and rheological properties of methacrylamide modified gelatin hydrogels. *Biomacromolecules* **1**, 31-38 (2000).
- [18] Kong B. et al. Fiber reinforced GelMA hydrogel to induce the regeneration of corneal stroma. *Nat. Commun.* **11**, 1-12 (2020).
- [19] Hua, M. et al. Strong tough hydrogels via the synergy of freeze-casting and salting out. *Nature* **590**, 594-599 (2021).
- [20] Yang, Q. et al. Photocurable bioresorbable adhesives as functional interfaces between flexible bioelectronic devices and soft biological tissues. *Nat. Mater.* **20**, 1559-1570 (2021).

Our modifications: We have modified the following sentence in the manuscript.

“However, most of them are far from practical and multipurpose uses, unlike their non-transient counterparts, due to insufficient characteristics, such as uncontrollable dissolution behaviors, limited mechanical elasticity/modulus/vulnerability, or poor storage stability.”

To

“However, most of them are far from practical and multipurpose uses, unlike their non-transient counterparts, due to insufficient characteristics, such as uncontrollable dissolution behaviors [1-6], limited mechanical elasticity/modulus/vulnerability [2-4,7-11], or poor storage stability [3-6,11] (Supplementary Table 1).”

- [1] Chen, S. et al. Mechanically and biologically skin-like elastomers for bio-integrated electronics. *Nat. Commun.* **11**, 1–8 (2020).
- [2] Baumgartner, M. et al. Resilient yet entirely degradable gelatin-based biogels for soft robots and electronics. *Nat. Mater.* **19**, 1102–1109 (2020).
- [3] Van Den Bulcke, A. I. et al. Structural and rheological properties of methacrylamide modified gelatin hydrogels. *Biomacromolecules* **1**, 31-38 (2000).
- [4] Kong B. et al. Fiber reinforced GelMA hydrogel to induce the regeneration of corneal stroma. *Nat. Commun.* **11**, 1-12 (2020).
- [5] Mutian, H. et al. Strong tough hydrogels via the synergy of freeze-casting and salting out. *Nature* **590**, 594-599 (2021).
- [6] Quansan, Y. et al. Photocurable bioresorbable adhesives as functional interfaces between flexible bioelectronic devices and soft biological tissues. *Nat. Mater.* **20**, 1559-1570 (2021).
- [7] Yang, J., Antonio R. W. & Guillermo A. A. Novel citric acid-based biodegradable elastomers for tissue engineering. *Adv. Mater.* **16**, 511-516 (2004).

- [8] Choi, Y. S. et al. Stretchable, dynamic covalent polymers for soft, long-lived bioresorbable electronic stimulators designed to facilitate neuromuscular regeneration. *Nat. Commun.* **11**, 1–14 (2020).
- [9] Wang, Yadong, et al. A tough biodegradable elastomer. *Nat. Biotechnol.* **20**, 602–606 (2002).
- [10] Held, M. et al. Soft electronic platforms combining elastomeric stretchability and biodegradability. *Adv. Sustain. Syst.* **6**, 2100035 (2022).
- [11] Pena-Francesch, A., Jung, H., Demirel, M. C. & Sitti, M. Biosynthetic self-healing materials for soft machines. *Nat. Mater.* **19**, 1230–1235 (2020).

Comment #3: Fig 2i, how about the comparison with biodegradable and stretchable hydrogels? For example, such as the following reported work, ref: Strong tough hydrogels via the synergy of freeze-casting and salting out. Nature 590, 594–599 (2021). <https://doi.org/10.1038/s41586-021-03212-z>; Photocurable bioresorbable adhesives as functional interfaces between flexible bioelectronic devices and soft biological tissues. Nat. Mater. 20, 1559–1570 (2021). <https://doi.org/10.1038/s41563-021-01051-x>.

Our response: Compared to typical polymers, there are some drawbacks of hydrogels, which are originated from the moisture-rich nature. First, it is difficult to integrate hydrogels with electronic components that physically dissolve or electrically degrade when in contact with water. Such issue hinders advanced/complex functions and a high level of performance. Second, when used in dry environments, their water content will gradually decrease by evaporation, finally causing physical deformation/delamination, thus, malfunction. Although some chemical/physical treatments could address these issues, such modifications may restrict fabrication procedures, limit processability and scalability. Nevertheless, hydrogels reported in recent works, including two papers the reviewer suggested and the paper we cited (*Nat. Mater.* **19**, 1230–1235 (2020)), exhibit good mechanical properties and proper degradability. We cited these papers to provide readers with more examples of biodegradable/stretchable polymers.

Our modifications: We have modified the following sentence and cited a few more papers, including the papers suggested by the reviewer.

“..... different types of biodegradable and stretchable materials based on protein, gelatin, and crosslinked polymers [1-5].”

- [1] Baumgartner, M. et al. Resilient yet entirely degradable gelatin-based biogels for soft robots and electronics. *Nat. Mater.* **19**, 1102–1109 (2020).
- [2] Pena-Francesch, A., Jung, H., Demirel, M. C. & Sitti, M. Biosynthetic self-healing materials for soft machines. *Nat. Mater.* **19**, 1230–1235 (2020).
- [3] Choi, Y. S. et al. Stretchable, dynamic covalent polymers for soft, long-lived bioresorbable electronic stimulators designed to facilitate neuromuscular regeneration. *Nat. Commun.* **11**, 1–14 (2020).

- [4] Chen, S. et al. Mechanically and biologically skin-like elastomers for bio-integrated electronics. *Nat. Commun.* **11**, 1–8 (2020).
- [5] Held, M. et al. Soft Electronic Platforms Combining Elastomeric Stretchability and Biodegradability. *Adv. Sustain. Syst.* **6**, 2100035 (2022).

To

“..... different types of biodegradable and stretchable materials based on protein [1], hydrogels [2-4], and crosslinked polymers [5-8].”

- [1] Pena-Francesch, A., Jung, H., Demirel, M. C. & Sitti, M. Biosynthetic self-healing materials for soft machines. *Nat. Mater.* **19**, 1230–1235 (2020).
- [2] Baumgartner, M. et al. Resilient yet entirely degradable gelatin-based biogels for soft robots and electronics. *Nat. Mater.* **19**, 1102–1109 (2020).
- [3] Kong, B. et al. Fiber reinforced GelMA hydrogel to induce the regeneration of corneal stroma. *Nat. Commun.* **11**, 1-12 (2020).
- [4] Hua, M. et al. Strong tough hydrogels via the synergy of freeze-casting and salting out. *Nature* **590**, 594-599 (2021).
- [5] Choi, Y. S. et al. Stretchable, dynamic covalent polymers for soft, long-lived bioresorbable electronic stimulators designed to facilitate neuromuscular regeneration. *Nat. Commun.* **11**, 1–14 (2020).
- [6] Chen, S. et al. Mechanically and biologically skin-like elastomers for bio-integrated electronics. *Nat. Commun.* **11**, 1–8 (2020).
- [7] Held, M. et al. Soft Electronic Platforms Combining Elastomeric Stretchability and Biodegradability. *Adv. Sustain. Syst.* **6**, 2100035 (2022).
- [8] Yang, Q. et al. Photocurable bioresorbable adhesives as functional interfaces between flexible bioelectronic devices and soft biological tissues. *Nat. Mater.* **20**, 1559-1570 (2021).

Comment #4: To achieve biodegradable conductors, the authors mixed PEDOT:PSS and ionic liquid with PLCL. PEDOT:PSS is not dissolvable in aqueous solutions, and ionic liquids are often cytotoxic. The authors need to provide detailed evidence and discussions, otherwise bioresorbable electronics based on these materials are not feasible.

Our response: As the reviewer commented, PEDOT:PSS is generally known as water-insoluble material. Nevertheless, there have been a few reports to elucidate degradation mechanism of PEDOT, i.e., hydrolysis by aqueous salt solutions [1] or oxidation by hydrogen peroxide that is a physiological oxidant [2,3]. Another study showed that macrophages cultured with PEDOT:PSS generated a significant amount of hydrogen peroxide that can initiate cellular phagocytosis [4]. Therefore, combined reactions of such hydrolysis and oxidation could possibly decompose PEDOT:PSS under physiological conditions, in-depth studies are still needed though.

According to the previous report [5], P14[TFSI] showed low toxicity toward primary human cells (similar to biocompatible choline-based ILs) and inherent biodegradability. Even considering that amount of the P14[TFSI] used in our composites is similar with that in toxicity tests of the previous report, ionic liquid-

induced cytotoxicity seemed to be negligible, as indirectly confirmed by histological examination and blood test for 8 weeks of device implantation (Figures 5f-h and Figures S31 and S32).

- [1] Sethumadhavan, V. et al. Hydrolysis of doped conducting polymers. *Commun. Chem.* **3**, 153-161 (2020).
- [2] Chen, T., Lin, Y., Bi, X. & Gu, Y. Conductive poly(3,4-ethylenedioxythiophene) is effectively degradable by hydrogen peroxide with iron (II) chloride. *Mater. Chem. Phys.* **242**, 122509-122513 (2020).
- [3] Thaning, E. M., Asplund, M. L. M., Nyberg, T. A., Inganas, O. W. & Holst, H. Stability of Poly(3,4-ethylenedioxythiophene) Materials Intended for Implants. *J. Biomed. Mater. Res.* **93B**, 407-415 (2010).
- [4] Gong, Hua. Stimulation of immune systems by conjugated polymers and their potential as an alternative vaccine adjuvant. *Nanoscale* **7**, 19282-19292 (2015).
- [5] Tang, B., Schneiderman, D. K., Bidoky, F. Z., Frisbie, C. D. & Lodge, T. P. Printable, Degradable, and Biocompatible Ion Gels from a Renewable ABA Triblock Polyester and a Low Toxicity Ionic Liquid. *ACS Macro Lett.* **6**, 1083-1088 (2017).

Our modifications: We have modified/added the following sentences and cited related references to the manuscript.

“..... (P14[TFSI]) as a biocompatible plasticizer.”

To

“..... (P14[TFSI]) as a biocompatible/biodegradable plasticizer [1].”

“..... with an appropriate encapsulation strategy (Supplementary Fig. 30).”

To

“..... with an appropriate encapsulation strategy (Supplementary Fig. 30). We couldn't observe substantial physical degradation of the device throughout the implantation period of 8 weeks due to the slow dissolution rate. However, we believe that constituting materials would gradually degrade as PLCL dissolves. Although chronic degradation of PEDOT:PSS in vivo has not been clearly unveiled yet, there are several reports that PEDOT degrades via hydrolysis in aqueous salt solutions [2], oxidation by hydrogen peroxide [3,4], or cellular phagocytosis by macrophage [5]. Therefore, we expect that combination of these reactions in physiological condition could possibly decompose PEDOT:PSS.”

- [1] Tang, B., Schneiderman, D. K., Bidoky, F. Z., Frisbie, C. D. & Lodge, T. P. Printable, Degradable, and Biocompatible Ion Gels from a Renewable ABA Triblock Polyester and a Low Toxicity Ionic Liquid. *ACS Macro Lett.* **6**, 1083-1088 (2017).
- [2] Sethumadhavan, V. et al. Hydrolysis of doped conducting polymers. *Commun. Chem.* **3**, 153-161 (2020).
- [3] Chen, T., Lin, Y., Bi, X. & Gu, Y. Conductive poly(3,4-ethylenedioxythiophene) is effectively degradable by hydrogen peroxide with iron (II) chloride. *Mater. Chem. Phys.* **242**, 122509-122513 (2020).
- [4] Thaning, E. M., Asplund, M. L. M., Nyberg, T. A., Inganas, O. W. & Holst, H. Stability of Poly(3,4-ethylenedioxythiophene) Materials Intended for Implants. *J. Biomed. Mater. Res.* **93B**, 407-415 (2010).

[5] Gong, H. et al. Stimulation of immune systems by conjugated polymers and their potential as an alternative vaccine adjuvant. *Nanoscale* **7**, 19282-19292 (2015).

Comment #5: Methodology regarding animal experiments for cardiac jacket are missing. For example, how were the devices deployed to the hearts of rats? How were the devices connected to external equipment for measurement over the implantation period? How was the connection protected from damage by the rats as biting or scratching of wire connectors often occur? Is the current device connection strategy feasible for clinical settings?

Our response: Figure S27 includes detailed images and approach of integration of cardiac jackets onto an artificial heart model. Briefly, we adjusted cardiac jackets into the 3D-designed shape prior to implantations, to facilitate system integration and to minimize any infection/malfunction of the heart. We also measured ECG signals while fastening the straps to find a proper force for better signal outputs and stable cardiac movements.

Connection strategy is described in the experimental section of the Supplementary Information. A pin connector wired to the cardiac jacket was left in the subcutaneous region of the animal model with pins exposed outside the rat chest. For measurement/stimulation weekly, the exposed pins were connected to external equipments after anesthetizing of the rat.

In order to protect the exposed pins from biting or scratching, we tailored a spandex jacket that was fitted to an animal model. We have added the following figure to the Supplementary Information.

Supplementary Figure 28. (a) Schematic illustration of device connection strategy. (b) Pins exposed outside the rat chest for connection with external equipment, where the pin connector wired to the cardiac jacket was left in the subcutaneous region of the rat. (c) A custom-made spandex jacket for protecting the pins, which allowed for stable measurement/stimulation during the implantation period of 8 weeks.

The current device connection strategy is limited for clinical use because of the issues that may arise from pin damage, pin connector-induced infection, and movement restriction during measurement/stimulation. To address these issues, our group has been working on a wireless communication and power transfer system. Together with this system, further implementation of integrated-circuit (IC) chips, a microcontroller unit (MCU), and Bluetooth modules can make our device feasible for clinical applications. This will be our future work.

Our modifications: We have added the following figure to the Supplementary Information.

Supplementary Figure 28. (a) Schematic illustration of device connection strategy. (b) Pins exposed outside the rat chest for connection with external equipment, where the pin connector wired to the cardiac jacket was left in the subcutaneous region of the rat. (c) A custom-made spandex jacket for protecting the pins, which allowed for stable measurement/stimulation during the implantation period of 8 weeks.

We have modified the following sentence in the manuscript.

“Integration process of the cardiac jacket on a 3D heart model appears in Supplementary Fig. 27.”

To

“Integration process of the cardiac jacket on a 3D heart model and the device connection strategy appear in Supplementary Figs. 27 and 28.”

Comment #6: Fig. 5a, strain sensors are involved in the device. Temperature changes indicating heart malfunctions can also affect the output signals of the sensor. How to differentiate these two parameters?

Our response: As the reviewer mentioned, the resistance of PLCL/Mo (9:1) composite increases as the temperature rises from 35 °C to 39 °C, primarily due to the decrease in conductivity of Mo flakes and the thermal expansion of PLCL matrix (See the left figure below). And, the PLCL/Mo composite showed a similar temperature-independent change in resistance ($R - R_0$), at 0.05 % strain generated by cardiac movements, which was estimated by comparison of Figures S15 and S30 (See the right figure below). Separation of these variables might be achieved through use of low thermal conductive materials and encapsulation with thermally-insulative materials. If a temperature sensor was additionally integrated into the cardiac jacket, the temperature change caused by heart malfunctions could be monitored and used for calibrating output signals.

Figure. Resistance changes of PLCL/Mo (9:1) composite (a) as a function of temperature and (b) at a strain of 0.05 % in a temperature range of 35 to 39 °C..

Our modifications: None.

Comment #7: The authors demonstrated ECG recording and electrical stimulation using the cardiac jacket. But it is not clear what clinical diagnostic functions or treatment the device can deliver. For example, the authors need to elaborate how can the device predict myocardial stiffness as claimed in the paper.

Our response: Although not discussed in depth in this study, the multi-functional cardiac jacket can be applied to diagnose and treat various heart diseases. In terms of diagnosis, arrhythmia, bradycardia, tachycardia, and myocardial infarction (MI) can be diagnosed by analyzing the frequency and waveform of ECG signals. For example,

changes in ECG signals, such as prolonged QT interval, ST segment changes, and T-wave abnormalities, are observed in hearts with cardiac fibrosis induced by myocardial infarction. Such cardiac fibrosis reduces the elasticity of myocardium, resulting in a decrease in ECG amplitude during the rhythmic contraction of the heart. In terms of treatment, electrical stimulation (pacing) can be used to treat ventricular tachycardia and ventricular fibrillation that appear spontaneously in the post-MI heart [1]. And, if electrical stimulation with low amplitude and high frequency is continuously applied to the post-MI heart, ventricular regeneration can be induced in a positive way [2-5].

A myocardial infarction, that occurred when blood flow decreases or stops to the coronary artery of the heart, results in damage of the myocardium, particularly in the left ventricle. This damage accompanies fibrosis generation, which increases the stiffness of cardiomyocytes and thus reduces the elasticity during systole/diastole (See the figure below [6]). Therefore, by monitoring the amplitude change of output signals detected by the strain sensor, we can predict the relative stiffness of post-MI heart (vs. normal heart). Although we monitored variations of the strain gauge for 2-3 weeks in this study, other encapsulation methods and design layouts will enable long-term observation for practical uses in clinical settings. We have provided the information about how the device can predict myocardial stiffness in the Supplementary Information.

Figure 2. Consequences of stiffness changes for cardiomyocytes. The increase of ventricular cardiac stiffness results in reduced diastolic blood volume. Blue arrows indicate the ventricular diastolic expansion. A stiffer matrix can interfere with sarcomere assembly and leads to changes in cardiomyocyte behavior by affecting cellular contraction and traction forces. Changes in matrix stiffness also cause aberrant gene expression that alters the balance of cardiac versus ECM genes

- [1] Park, J. et al. Electromechanical cardioplasty using a wrapped elasto-conductive epicardial mesh. *Sci. Transl. Med.* **8**, 344ra86-344ra86 (2016).
- [2] Mukherjee, R. et al. Long-term localized high-frequency electric stimulation within the myocardial infarct: effects on matrix metalloproteinases and regional remodeling. *Circulation* **122**, 20-32 (2010).
- [3] Spadaccio, C. et al. In situ electrostimulation drives a regenerative shift in the zone of infarcted myocardium. *Cell transplantation* **22**, 493-503, (2013).
- [4] Uitterdijk, A. et al. Intermittent pacing therapy favorably modulates infarct remodeling. *Basic Research in Cardiology* **112**, 1-10, (2017).

[5] Genau, M. C. et al. Institution of localized high-frequency electrical stimulation targeting early myocardial infarction: Effects on left ventricle function and geometry. *The Journal of Thoracic and Cardiovascular Surgery* **156**, 568-575, (2018).

[6] Münch, J. & Abdelilah-Seyfried, S. Sensing and responding of cardiomyocytes to changes of tissue stiffness in the diseased heart. *Frontiers in Cell and Developmental Biology* **9**, 1-13, (2021).

Our modifications: We have modified the following sentence in the manuscript and the Supplementary Figure caption.

“..... a non-invasive medical implant that can accomplish sensing/therapeutic functions for various cardiac diseases over long periods of time.”

To

“..... a non-invasive medical implant that can accomplish sensing/therapeutic functions for various cardiac diseases, such as arrhythmia, tachycardia, fibrillation, and myocardial infarction, over long periods of time.”

“**Supplementary Figure 30.** Monitoring of myocardial strain with the PLCL/Mo sensor on the implanted device for 4 weeks to predict cardiac mechanics in early stages of heart diseases.”

To

“**Supplementary Figure 30.** Monitoring of myocardial strain with the PLCL/Mo sensor on the implanted device for 4 weeks to predict cardiac mechanics in the early stages of heart diseases. A myocardial infarction, that occurred when blood flow decreases or stops to the coronary artery of the heart, results in damage of the myocardium, particularly in the left ventricle. This damage accompanies fibrosis generation, which increases the stiffness of cardiomyocytes and thus reduces the elasticity during systole/diastole. Therefore, by monitoring amplitude change of output signals detected by strain sensor, we can predict the relative stiffness of post-MI heart (vs. normal heart).”

Comment #8: Fig. 3g, the authors stated that “the stretchability and conductivity far surpass results in previous reports”, which seems not true as the reported stretchability of ref 46 looks better than the current work.

Our response: We thank the reviewer for this comment. We liked to highlight the overall performance of PLCL/PP rather than just values of stretchability and conductivity. Although the stretchability of the composite in reference 46 exhibits ~200 % higher than the one of PLCL/PP, the conductivity (< 0.005 S/cm) is ~5 orders of magnitude lower than that of PLCL/PP (~200 S/cm) and is even not in a metallic range. We have modified the manuscript to avoid misreading.

Our modifications: We have modified the following sentence.

"..... the overall performance in both conductivity (>200 S/cm) and stretchability (>500 %) far surpassed those of degradable composites"

To

"..... the overall performance (conductivity, >200 S/cm; stretchability, >500 %) as elastic conductor far surpassed those of degradable composites"

Comment #9: Other issues, in the methodology part, Mo films ~ 500 micrometers were sputtered onto PMMA. Is sputtering such thick Mo films a scalable process?

Our response: We thank the reviewer for noticing this typo.

Our modifications: We replaced '500 μm -thick' with '500 nm-thick'.

Reviewer #2 (Remarks to the Author):

In this manuscript, a highly stretchable and biodegradable elastomer is synthesized as the substrate for soft transient electronics. Biodegradable transient devices show promising potentials in implantable bioelectronics and environmental-friendly electronics. A series of eye-catching demonstrations are presented in the manuscript to illustrate the potential applications of the elastomer. Despite the interesting aspects, I find several major inconsistencies in the study involving non-degradable/partially degradable devices. If the new property is not harnessed, the cool demonstrations may not be very relevant. Tremendous work is likely required to fix many scientific/technical issues. Otherwise, the study may not qualify for this prestigious journal.

Our response: We appreciate the reviewer's valuable comments. We believe that the manuscript improved by addressing all the scientific/technical issues raised from the reviewer will qualify for publishing in *Nature Communications*.

Comment #1: An LED array is the first device demonstration. Most electronic components are non-degradable materials, including PMMA, PI, Surface mount chips, silver epoxy, etc. I am surprised that the demonstration is standard electronics on a biodegradable substrate. I cannot find any major benefits of this device system for practical applications.

Our response: We thank the reviewer for this assessment. In this demonstration, we intended to highlight outstanding mechanical/physical properties of PLCL, such as stretchability, toughness, adhesion, transparency, and chemical resistance, which are essential for substrate/encapsulation of versatile electronic devices that can work in various deformation modes. Therefore, we chose the large-scale, complex, and individually addressable LED array as a representative example. For completely dissolvable form, we constructed a complementary-metal-oxide-semiconductor (CMOS) inverter array that can operate properly at a tensile strain of 200 %, as shown in Figure S9.

Supplementary Figure 9. Fully-dissolvable, stretchable complementary metal-oxide-semiconductor (CMOS) inverter array on PLCL. (a,b) Schematic exploded view (a) and optical image (b) of the CMOS array on PLCL substrate. (c) Output voltage characteristics of a representative CMOS inverter at $V_{dd} = 10$ V. The gain was ~ 60 . (d) Linear (black) and log scale (blue) transfer curves of a typical p-channel metal-oxide-semiconductor field-effect transistor (MOSFET). The channel length and width were $5 \mu\text{m}$ and $300 \mu\text{m}$, respectively. The mobility (linear regime) and on/off ratio were $\sim 78 \text{ cm}^2/\text{V}\cdot\text{s}$ and ~ 106 , respectively. (e) I-V characteristics of a typical n-channel MOSFET. The channel length and width were $20 \mu\text{m}$ and $100 \mu\text{m}$, respectively. The mobility (linear regime) and on/off ratio were $\sim 310 \text{ cm}^2/\text{V}\cdot\text{s}$ and ~ 105 , respectively. (e) Optical images and the corresponding mechanical simulations of the CMOS with (left) and without (right) a uniaxial strain of 200%. (f) Output voltage characteristics of a CMOS inverter ($V_{dd} = 10$ V) with tensile strains of 0, 100, and 200%.

Our modifications: None.

Comment #2: A conductive composite is prepared as a mixture of PEDOT:PSS, P14[TFSI], and PLCL. PEDOT:PSS is a well-recognized biocompatible polymer rather than a biodegradable one. The degradability of the plasticizer is also open to question. With limited biodegradability, the conductive composite is better considered a transient conductor.

Our response: As the reviewer commented, PEDOT:PSS is generally known as water-insoluble material. Nevertheless, there have been a few reports to elucidate degradation mechanism of PEDOT, such as hydrolysis by aqueous salt solutions [1] or oxidation by hydrogen peroxide, which is a physiological oxidant [2,3]. Another study showed that macrophages cultured with PEDOT:PSS generated a significant amount of hydrogen peroxide that can initiate cellular phagocytosis [4]. Therefore, combinatorial reactions of such hydrolysis and oxidation could possibly decompose PEDOT:PSS in physiological condition, in-depth studies are still needed though.

According to the previous report [5], P14[TFSI] showed low toxicity toward primary human cells (similar to biocompatible choline-based ILs) and inherent biodegradability. Even considering that amount of the P14[TFSI] used in our composites is similar with that in toxicity tests of the previous report, ionic liquid-induced cytotoxicity seemed to be negligible, as indirectly confirmed by histological examination and blood test for 8 weeks of device implantation (Figures 5f-h and Figures S31 and S32).

[1] Sethumadhavan, V. et al. Hydrolysis of doped conducting polymers. *Commun. Chem.* **3**, 153-161 (2020).

[2] Chen, T., Lin, Y., Bi, X. & Gu, Y. Conductive poly(3,4-ethylenedioxythiophene) is effectively degradable by hydrogen peroxide with iron (II) chloride. *Mater. Chem. Phys.* **242**, 122509-122513 (2020).

[3] Thaning, E. M., Asplund, M. L. M., Nyberg, T. A., Inganas, O. W. & Holst, H. Stability of Poly(3,4-ethylenedioxythiophene) Materials Intended for Implants. *J. Biomed. Mater. Res.* **93B**, 407-415 (2010).

[4] Gong, Hua. Stimulation of immune systems by conjugated polymers and their potential as an alternative vaccine adjuvant. *Nanoscale* **7**, 19282-19292 (2015).

[5] Tang, B., Schneiderman, D. K., Bidoky, F. Z., Frisbie, C. D. & Lodge, T. P. Printable, Degradable, and Biocompatible Ion Gels from a Renewable ABA Triblock Polyester and a Low Toxicity Ionic Liquid. *ACS Macro Lett.* **6**, 1083-1088 (2017).

Our modifications: We have modified/added the following sentences and cited related references to the manuscript.

“..... (P14[TFSI]) as a biocompatible plasticizer.”

To

“..... (P14[TFSI]) as a biocompatible/biodegradable plasticizer [1].”

“..... with an appropriate encapsulation strategy (Supplementary Fig. 30).”

To

“..... with an appropriate encapsulation strategy (Supplementary Fig. 30). We couldn't observe substantial physical degradation of the device throughout the implantation period of 8 weeks due to the slow dissolution rate. However, we believe

that constituting materials would gradually degrade as PLCL dissolves. Although chronic degradation of PEDOT:PSS in vivo has not been clearly unveiled yet, there are several reports that PEDOT degrades via hydrolysis in aqueous salt solutions [2], oxidation by hydrogen peroxide [3,4], or cellular phagocytosis by macrophage [5]. Therefore, we expect that combination of these reactions in physiological condition could possibly decompose PEDOT:PSS.”

[1] Tang, B., Schneiderman, D. K., Bidoky, F. Z., Frisbie, C. D. & Lodge, T. P. Printable, Degradable, and Biocompatible Ion Gels from a Renewable ABA Triblock Polyester and a Low Toxicity Ionic Liquid. *ACS Macro Lett.* **6**, 1083-1088 (2017).

[2] Sethumadhavan, V. et al. Hydrolysis of doped conducting polymers. *Commun. Chem.* **3**, 153-161 (2020).

[3] Chen, T., Lin, Y., Bi, X. & Gu, Y. Conductive poly(3,4-ethylenedioxythiophene) is effectively degradable by hydrogen peroxide with iron (II) chloride. *Mater. Chem. Phys.* **242**, 122509-122513 (2020).

[4] Thaning, E. M., Asplund, M. L. M., Nyberg, T. A., Inganas, O. W. & Holst, H. Stability of Poly(3,4-ethylenedioxythiophene) Materials Intended for Implants. *J. Biomed. Mater. Res.* **93B**, 407-415 (2010).

[5] Gong, Hua. Stimulation of immune systems by conjugated polymers and their potential as an alternative vaccine adjuvant. *Nanoscale* **7**, 19282-19292 (2015).

Comment #3: The conductive composite is used to construct an electronic mesh for cardiac monitoring and stimulation. Additional experiments deal with biocompatibility. Similar results are easily obtained with standard silicone (e.g., Ecoflex). The in vivo experiments have nothing to do with the biodegradable/transient characteristics of the elastomer.

Our response: We thank the reviewer for this comment. The use of well-known silicon elastomers, such as Ecoflex and PDMS, can enable the fabrication of an electronic mesh and corresponding electronic components for cardiac monitoring/stimulation. However, as presented by previous studies on bioresorbable electronics [1-3], the biodegradable nature of these electronics imparts many advantages over traditional electronics in the context of biomedical applications, particularly for implantable devices. The primary benefit is the elimination of the need for a second surgery to retrieve the device, as biodegradable electronics are designed to degrade/dissolve spontaneously in the body over time after completing intended functions. This feature helps minimize the risk of complications associated with a second surgery, such as inflammation, fibrosis, and infection. Notably, the heart is one of the most vital and sensitive organs of the human body, making it imperative to avoid frequent surgery that may lead to adverse outcomes, such as blood clots, arrhythmias, stroke, heart attack, and even death. Furthermore, the elimination of a second surgery can make recovery times shorter, reduce the overall cost of treatment for patients, and alleviates the effort required of medical staff.

[1] Choi, Y. S. et al. Fully implantable and bioresorbable cardiac pacemakers without leads or batteries. *Nat. Biotechnol.* **39**, 1228–1238 (2021).

[2] Kang, S. -K. et al. Bioresorbable silicon electronic sensors for the brain. *Nature* **530**, 71-79 (2016).

[3] Yu, K. J. et al. Bioresorbable silicon electronics for transient spatiotemporal mapping of electrical activity from the cerebral cortex. *Nat. Mater.* **15**, 782-792, (2016).

Our modifications: None.

Comment #4: The biodegradable elastomer has not been effectively characterized with benign reaction products suitable for implantable applications. The experimental tests actually deal with environmental degradation.

Our response: Along with the environmental degradation of PLCL by fungi (Figures 1c–e), we also evaluated the dissolution of PLCLs with various Mn in artificial cerebrospinal fluid (ACSF) and phosphate-buffered saline (PBS) to investigate degradation behaviors under in vivo environments (Figures 2a and 2b and Figure S3). Cytocompatibility of by-products of PLCL has been already reported in previous research [1]. When exposed to water, PLCL, a polyester, decomposes into lactic and caproic acids via hydrolysis of ester bonds, causing a decrease in pH level; however, such change is negligible to induce other effects on physiological functions, including homeostasis [2]. This report was indirectly confirmed by our results, such as histological examination, blood test, and body weight tracking, for 8 weeks of device implantation (Figures 5f-h and Figures S31 and S32), indicating the suitability of PLCL for implantable applications. For better understanding, we have provided additional information to the manuscript.

[1] Jeong, S. I., et al. In vivo biocompatibility and degradation behavior of elastic poly (l-lactide-co- ϵ -caprolactone) scaffolds. *Biomaterials* **25**, 5939-5946 (2004).

[2] Sabbatier, G., et al. Design, Degradation Mechanism and Long-Term Cytotoxicity of Poly (l-lactide) and Poly (Lactide-co- ϵ -Caprolactone) Terpolymer Film and Air-Spun Nanofiber Scaffold. *Macromol. Biosci.* **15**, 1392-1410 (2015).

Our modifications: We have modified the following sentence and added related references.

“PLCL films (140k) dissolved via hydrolysis over time and”

To

“PLCL films (140k) dissolved via hydrolysis over time without remaining any toxic by-products [1,2] and”

[1] Jeong, S. I. et al. In vivo biocompatibility and degradation behavior of elastic poly (l-lactide-co- ϵ -caprolactone) scaffolds. *Biomaterials* **25**, 5939-5946 (2004).

[2] Sabbatier, G. et al. Design, Degradation Mechanism and Long-Term Cytotoxicity of Poly (l-lactide) and Poly (Lactide-co- ϵ -Caprolactone) Terpolymer Film and Air-Spun Nanofiber Scaffold. *Macromol. Biosci.* **15**, 1392-1410 (2015).

Comment #5; Poly(L-lactide-co- ϵ -caprolactone) (PLCL) is widely used in tissue engineering with limited stretchability. The underlying mechanism of the improved stretchability observed here should be further clarified. In design and synthesis, the differences from previous reports should also be explicitly explained.

Our response: We thank the reviewer for the detailed comment. As stated in the manuscript, L-lactide (LA) and ϵ -caprolactone (CL) play a role of hard and soft domains in PLCL, respectively, and physical crosslinking of such domains yield the elasticity. Here, the higher the ratio of LA to CL, the higher the mechanical modulus but the lower the stretchability, and vice versa. Other conditions on chemical synthesis, including types of initiator, monomers to initiator ratio, temperature, and humidity, affect the mechanical properties. Consequently, previous works using a high LA to CL ratio [1,3], multi-ol initiator [2,5], high monomers to initiator ratio [2,4,5], and low temperature [1] produced PLCL elastomers with low stretchability. In this respect, since we synthesized PLCLs with 5:5 of LA to CL ratio by using '1-dodecanol' as an initiator in this work, the PLCLs exhibited outstanding stretchability up to $\sim 1600\%$ as well as softness, as consistent with previous results (PLCLs with 5:5 ratio show $\sim 2000\%$ in stretchability [3]). Furthermore, we diversified the mechanical properties of PLCL by controlling M_n from 60k to 200k as shown in Figure 2d. For better understanding, we have added more information to the manuscript.

[1] Fernandez, Jorge., Etxeberria, Agustin. & Sarasua, Jose-Ramon. Synthesis, structure and properties of poly(L-lactide-co- ϵ -caprolactone) statistical copolymers. *J. Mech. Behav. Biomed. Mater.* **9**, 100–112 (2012).

[2] Lee, H. S. et al. Development of a regenerative porous PLCL nerve guidance conduit with swellable hydrogel-based microgrooved surface pattern via 3D printing. *Acta Biomater.* **141**, 219-232 (2022).

[3] Zhang, M., Chang, Z., Wang, X. & Li, Qian. Synthesis of Poly(L-lactide-co- ϵ -caprolactone) Copolymer: Structure, Toughness, and Elasticity. *Polymers* **13**, 1270-1282 (2021).

[4] Jeong, S. I. et al. Manufacture of elastic biodegradable PLCL scaffolds for mechano-active vascular tissue engineering. *J. Biomater. Sci.* **15** 645-660 (2004).

[5] Jeong, S. I. et al. Morphology of Elastic Poly(L-lactide-co- ϵ -caprolactone) Copolymers and in Vitro and in Vivo Degradation Behavior of Their Scaffolds. *Biomacromolecules.* **5**, 1303-1309 (2004).

Our modifications: We have modified the following sentence in the manuscript.

“..... where physical cross-linking of hard (LA) and soft domains (CL) contributes polymer elasticity (Supplementary Fig. 1). Varying monomer to initiator ratios yields

different molecular weights (M_n) of PLCL elastomers (Supplementary Table 1), allowing for tunable chemical/physical/mechanical properties.”

To

“..... where physical cross-linking of hard (LA) and soft domains (CL) in a 5:5 ratio provides higher polymer elasticity and softness than previous works using high LA fractions [1,2] (Supplementary Fig. 1). Varying monomer to initiator ratios yields different molecular weights (M_n) of PLCL elastomers (Supplementary Table 1), allowing for tunable chemical/physical/mechanical properties.”

[1] Fernandez, Jorge., Etxeberria, Agustin. & Sarasua, Jose-Ramon. Synthesis, structure and properties of poly(L-lactide-co- ϵ -caprolactone) statistical copolymers. *J. Mech. Behav. Biomed. Mater.* **9**, 100–112 (2012).

[2] Zhang, M., Chang, Z., Wang, X. & Li, Qian. Synthesis of Poly(L-lactide-co- ϵ -caprolactone) Copolymer: Structure, Toughness, and Elasticity. *Polymers* **13**, 1270-1282 (2021).

Comment #6: The reaction mechanism of the degradation process should be explicitly explained to justify the products' benign nature.

Our response: When exposed to water, PLCL, a polyester, decomposes into lactic and caproic acids via hydrolysis of ester bonds, causing a decrease in pH level, however, such change is very negligible to induce negative effects on physiological functions, including homeostasis [1], as indirectly confirmed by our results, such as histological examination, blood test, and body weight tracking, for 8 weeks of device implantation (Figures 5f-h and Figures S31 and S32). In addition, cytocompatibility of lactic and caproic acids have been verified in many previous reports about PLA and PCL, which are commercially available representative biodegradable/biocompatible polymers [2]. For better understanding, we have added a schematic illustration describing the degradation process of PLCL to the manuscript.

[1] Sabbatier, G., et al. Design, Degradation Mechanism and Long-Term Cytotoxicity of Poly (l-lactide) and Poly (Lactide-co- ϵ -Caprolactone) Terpolymer Film and Air-Spun Nanofiber Scaffold. *Macromol. Biosci.* **15**, 1392-1410 (2015).

[2] Leja, K. & Lewandowicz, G. Polymer biodegradation and biodegradable polymers-a review. *Pol. J. Environ. Stud.* **19**, 255-266 (2010).

Our modifications: We have modified the following figure and added related references in the Supplementary information.

Supplementary Figure 3. Time-dependent changes in weight ratio of PLCL film (140k, 100 μm -thick) as a function of type (a), temperature (b), and pH levels (c) of solution. Variation in an ionic content/concentration affected the hydrolytic reaction, which caused rapid dissolution rates in artificial cerebrospinal fluid (ACSF) than deionized water, while high temperature and pH values accelerated the dissolution of PLCL.

To

Supplementary Figure 3. (a) Hydrolytic degradation mechanism of PLCL, where chain cleavage of ester bonds via hydrolysis generates biocompatible lactic and caproic acids (carboxylic acid) as final products [1,2]. (b-d) Time-dependent changes in weight ratio of PLCL film (140k, 100 μm -thick) as a function of type (b), temperature (c), and pH levels (d) of solution. Variation in an ionic content/concentration affected the hydrolytic reaction, which caused rapid dissolution rates in artificial cerebrospinal fluid (ACSF) than deionized water, while high temperature and pH values accelerated the dissolution of PLCL.

[1] Sabbatier, G., et al. Design, Degradation Mechanism and Long-Term Cytotoxicity of Poly (l-lactide) and Poly (Lactide-co- ϵ -Caprolactone) Terpolymer Film and Air-Spun Nanofiber Scaffold. *Macromol. Biosci.* **15**, 1392-1410 (2015).

[2] Leja, K. & Lewandowicz, G. Polymer biodegradation and biodegradable polymers-a review. *Pol. J. Environ. Stud.* **19**, 255-266 (2010).

Comment #7. It is better to explain the conductivity calculation of stretched conductive composites. The claim on strain-insensitive characteristics lacks any experimental evidence. In Fig 3d, the emission intensity drops in response to large tensile deformations. The device is obviously very sensitive to strains.

Our response: As stated in ‘Methods’ section, we calculated the electrical conductivity of conductive composites by the equation: electrical conductivity (S/cm) = $1 / (\text{sheet resistance (ohm/sq)} \times \text{thickness (cm)})$, where the sheet resistance and the thickness were defined by a four-point probe system (M4P205, MStech, South Korea) and SEM analysis, respectively. The same method was applied to stretched samples. As shown in Figure 3g, previous biodegradable conductive composites have somewhat insufficient elasticity or conductivity, while PLCL/PP relatively exhibited high stretchability and conductivity at the same time, suitable for an elastic conductor. This feature is what we would like to intend to highlight. To avoid misunderstanding, we have modified ‘strain-insensitive’ to ‘strain-tolerant’ throughout the manuscript.

Our modifications: We have modified the following sentences in the manuscript.

“Conductive elastomers insensitive/sensitive to mechanical deformations

To

“Conductive elastomers tolerant/sensitive to mechanical deformations

“..... particularly the strain-insensitive composites retain

To

“..... particularly the strain-tolerant composites retain

We have modified the following sentence in the Supplementary Information.

“..... and strain-insensitive PLCL/PP for

To

“..... and strain-tolerant PLCL/PP for

Comment #8: A PLCL/Mo composite is employed for strain sensing. Within a minor strain of 2%, the resistance increases by one order of magnitude over several hundred cycles. The significant drift contradicts the claim of reliable cyclic behaviors (Line 175). In addition, the limited cycling stability is a practical issue for cardiac strain monitoring involving excessive cycles. In Fig S30, the responses show obvious deteriorations over time.

Our response: We agree with the reviewer’s comment that the resistance change of PLCL/Mo composite was considerable at the initial stage of cyclic stability test. For clarity, we have removed the term ‘reliable cyclic behaviors’ from the manuscript.

Comparing the signal amplitudes that appeared in Figures S15 ($\Delta R/R_0$, ~ 100 ; cyclic test with 2 % strain) and S30 ($\Delta R/R_0$, ~ 0.04 ; cardiac strain monitoring), the magnitude of strain induced by cardiac movements was much lower than 2 % and was estimated to be approximately 0.05 %. At a strain of 0.05 %, PLCL/Mo shows stable cyclic behavior as shown in the figure below. Therefore, the time-dependent signal deteriorations in Figure S30 were presumably derived from the biodegradation of PLCL/Mo rather than cyclic instability, which can be addressed by other encapsulation strategies as a followup study.

Figure. Cyclic stability of PLCL/Mo at a tensile strain of 0.05 %

Our modifications: We have modified the following sentence in the manuscript.

“..... and negligible hysteresis, reliable cyclic behaviors, and particularly”

To

“..... and negligible hysteresis and particularly”

Reviewer #3 (Remarks to the Author):

The authors use poly(l-lactide-co-ε-caprolactone) (PLCL) as an elastomer for transient electronics. They demonstrate in detail a variety of device work and some degradation studies. Figures are well made and the experiments are thorough. I see the major advance of this paper is fabricating electronics on PLCL, which has not been done before. However, I do not consider PLCL a new elastomer system. That said, I believe this work merits publication in nature communications once the comments below are addressed.

Our response: We thank the reviewer for these positive comments and for the recommendation to publish in *Nature Communications*. We believe that the improved manuscript, through feedback on the reviewer's valuable comments, will inspire the journal community and readers.

Comment #1: In the introduction, the authors mention that “recent research efforts have introduced a few different types of biodegradable and stretchable materials based on protein, gelatin, and crosslinked polymers”. I recognize the authors attempt to highlight that there are limited polymers used for transient electronics, but it gives the impression that there is little work in this area for non-experts not familiar with the polymer field. There is rich literature on biodegradable elastomers, and detailed analysis of degradation. PLCL has been heavily studied, and there are studies of its use as substrates for electronic materials—people have blended it with inorganic/organic materials and even co-polymerized conductive polymers on its substrate. Often, researchers outside of the electronics-focused or applications-focused community omit these perspectives; I think it would be powerful to acknowledge these works.

Our response: We agree with the reviewer's comment that biodegradable elastomers were insufficiently introduced in this manuscript. To address this issue, we have made a table for introducing more examples of biodegradable polymers with their mechanical/biochemical properties and cited a few more papers in the manuscript.

We have found recent researches about the use of PLCL as a substrate for electronic materials [1-3] and cited these papers in the manuscript.

[1] Cheng, S. et al. Electronic Blood Vessel. *Matter* **3**, 1664-1684 (2020).

[2] Wang, Q. et al. Shape memory performances of homogeneous poly (L-lactide-co-ε-caprolactone)/polytrimethylene carbonate-grafted functionalized graphene oxide nanocomposites. *Eur. Polym. J.* **173**, 111291 (2022).

[3] Lu, S. et al. Polydopamine-Decorated PLCL Conduit to Induce Synergetic Effect of Electrical Stimulation and Topological Morphology for Peripheral Nerve Regeneration. *Small Methods* (2023) doi.org/10.1002/smt.202200883.

Our modifications: We have added the following table to the Supplementary Information.

Materials	T _g [°C]	T _m [°C]	Young's modulus [E, MPa]	Elongation at break [ε _b , %]	Degradation rate [weight loss in time]	Shelf life [in ambient conditions]	Issues	Ref	
PLCL (50/50)	-7 - -3	-	2.5 - 7.5	900 - 1600	20 % in 30 weeks	> 6 month		Our works	
Silk	-	-	10000	20	35 % in 3 weeks	-	High modulus Poor stretchability	[1]	
PLA	55	183	3000 - 3500	5 - 30	several years	5 years	High modulus Poor stretchability	[2,3]	
PCL	-60	60-62	297 - 363	700-1000	several years	5 years	High modulus Poor elasticity	[2,4,5]	
Biodegradable polymers	PGA	35-45	220	7000 - 8400	30	6 - 12 month	5 years	High modulus Poor stretchability	[4]
	PLGA (50/50)	40-60	153	2000	3 - 10	1 - 2 weeks	4 weeks in dry condition	High modulus Poor stretchability Rapid degradation	[3,6]
	PBS	5	163	700	6	10 % in 14 weeks	-	High modulus Poor stretchability	[7,8]
	PHB	5	180	3500	5-8	60 % in 24 weeks	-	High modulus Poor stretchability	[9]
	POC	0.2	-	0.92 - 16.4	260	50 % in 25 weeks	-	Low stretchability	[10]
	b-DCPU	-35 - -30	-	0.5 - 3.8	230	20 % in 10 weeks	-	Low stretchability	[11]
	PGS	-51.24, -18.50	5.23, 37.62	0.5	330	17 % in 60 days	-	Low stretchability	[12]
	PSeD-U	-20 - -2	-	0.54 - 0.74	136 - 509	60% in 8 hrs	-	Rapid degradation	[13]
	PGSA	-	-	0.143 - 0.592	20 - 130	90 % in 8 weeks	12 months	Low stretchability	[14]
	Gelatin biogel	-	-	0.03 - 0.3	320	60 % in 4 days	13 months	Low stretchability Rapid degradation	[15]
Hydrogels	TRn11	-	-	2.4	300	on demand	-	Low stretchability Poor storage stability	[16]
	GelMa	-	-	0.01 – few tens	150 - 290	10 days	6 months	Low stretchability Rapid degradation Poor storage stability	[17,18]
	HA-PVA	-	200-250	2.5	2900	N/A	-	Poor degradation Poor storage stability	[19]
	BTIM	-	-	0.03	1200	20 days	-	Rapid degradation Poor storage stability	[20]

Table S1. Summary of representative biodegradable polymers and hydrogels with their mechanical/degradation properties

[1] Vepari, C. & Kaplan, D. L. Silk as a biomaterial. *Prog. Polym. Sci.* **32**, 991–1007 (2007).

[2] Sarazin, P., Roy, X. & Favis, B. D. Controlled preparation and properties of porous poly (L-lactide) obtained from a co-continuous blend of two biodegradable polymers. *Biomaterials* **25**, 5965-5978 (2004).

[3] Gentile, P. et al. An Overview of Poly(lactic-co-glycolic) Acid (PLGA)-Based Biomaterials for Bone Tissue Engineering. *Int. J. Mol. Sci.* **15**, 3640-3659 (2014).

[4] Manavitehrani, I. et al. Biomedical Applications of Biodegradable Polyesters. *Polymers* **8**, 20 (2016).

[5] Eshraghi, S. & Das, S. Mechanical and microstructural properties of polycaprolactone scaffolds with one-dimensional, two-dimensional, and three-dimensional orthogonally oriented porous architectures produced by selective laser sintering. *Acta Biomater.* **6**, 2467-2476 (2010).

[6] Houchin, M. L., & E. M. Topp. Physical Properties of PLGA Films During Polymer Degradation. *J. Appl. Polym. Sci.* **114**, 2848-2854 (2009).

[7] Lindström, A., Albertsson, AC. & Hakkarainen, M. Quantitative determination of degradation products an effective means to study early stages of degradation in linear and branched poly (butylene adipate) and poly (butylene succinate). *Polym. Degrad. Stab.* **83**, 487-493 (2004).

- [8] Liu, L. et al. Mechanical properties of poly (butylene succinate) (PBS) biocomposites reinforced with surface modified jute fibre. *Compos. - A: Appl. Sci. Manuf.* **40**, 669-674 (2009).
- [9] Volova, T. et al. Results of biomedical investigations of PHB and PHB/PHV fibers. *Biochem. Eng. J.* **16**, 125-133 (2003).
- [10] Yang, J., Webb, A. R. & Ameer, G. A. Novel citric acid-based biodegradable elastomers for tissue engineering. *Adv. Mater.* **16**, 511-516 (2004).
- [11] Choi, Y. S. et al. Stretchable, dynamic covalent polymers for soft, long-lived bioresorbable electronic stimulators designed to facilitate neuromuscular regeneration. *Nat. Commun.* **11**, 1–14 (2020).
- [12] Wang, Y. et al. Yadong, et al. A tough biodegradable elastomer. *Nat. Biotechnol.* **20**, 602-606 (2002).
- [13] Chen, S. et al. Mechanically and biologically skin-like elastomers for bio-integrated electronics. *Nat. Commun.* **11**, 1–8 (2020).
- [14] Held, M. et al. Soft electronic platforms combining elastomeric stretchability and biodegradability. *Adv. Sustain. Syst.* **6**, 2100035 (2022).
- [15] Baumgartner, M. et al. Resilient yet entirely degradable gelatin-based biogels for soft robots and electronics. *Nat. Mater.* **19**, 1102–1109 (2020).
- [16] Pena-Francesch, A., Jung, H., Demirel, M. C. & Sitti, M. Biosynthetic self-healing materials for soft machines. *Nat. Mater.* **19**, 1230–1235 (2020).
- [17] Van Den Bulcke, A. I. et al. Structural and rheological properties of methacrylamide modified gelatin hydrogels. *Biomacromolecules* **1**, 31-38 (2000).
- [18] Kong, B. et al. Fiber reinforced GelMA hydrogel to induce the regeneration of corneal stroma. *Nat. Commun.* **11**, 1-12 (2020).
- [19] Hua, M. et al. Strong tough hydrogels via the synergy of freeze-casting and salting out. *Nature* **590**, 594-599 (2021).
- [20] Yang, Q. et al. Photocurable bioresorbable adhesives as functional interfaces between flexible bioelectronic devices and soft biological tissues. *Nat. Mater.* **20**, 1559-1570 (2021).

We have modified the following sentence and cited a few more papers.

“..... different types of biodegradable and stretchable materials based on protein, gelatin, and crosslinked polymers [1-5].”

- [1] Baumgartner, M. et al. Resilient yet entirely degradable gelatin-based biogels for soft robots and electronics. *Nat. Mater.* **19**, 1102–1109 (2020).
- [2] Pena-Francesch, A., Jung, H., Demirel, M. C. & Sitti, M. Biosynthetic self-healing materials for soft machines. *Nat. Mater.* **19**, 1230–1235 (2020).
- [3] Choi, Y. S. et al. Stretchable, dynamic covalent polymers for soft, long-lived bioresorbable electronic stimulators designed to facilitate neuromuscular regeneration. *Nat. Commun.* **11**, 1–14 (2020).
- [4] Chen, S. et al. Mechanically and biologically skin-like elastomers for bio-integrated electronics. *Nat. Commun.* **11**, 1–8 (2020).
- [5] Held, M. et al. Soft Electronic Platforms Combining Elastomeric Stretchability and Biodegradability. *Adv. Sustain. Syst.* **6**, 2100035 (2022).

To

“..... different types of biodegradable and stretchable materials based on protein [1], hydrogels [2-4], and crosslinked polymers [5-8].”

- [1] Pena-Francesch, A., Jung, H., Demirel, M. C. & Sitti, M. Biosynthetic self-healing materials for soft machines. *Nat. Mater.* **19**, 1230–1235 (2020).
- [2] Baumgartner, M. et al. Resilient yet entirely degradable gelatin-based biogels for soft robots and electronics. *Nat. Mater.* **19**, 1102–1109 (2020).
- [3] Kong, B. et al. Fiber reinforced GelMA hydrogel to induce the regeneration of corneal stroma. *Nat. Commun.* **11**, 1-12 (2020).
- [4] Hua, M. et al. Strong tough hydrogels via the synergy of freeze-casting and salting out. *Nature* **590**, 594-599 (2021).
- [5] Choi, Y. S. et al. Stretchable, dynamic covalent polymers for soft, long-lived bioresorbable electronic stimulators designed to facilitate neuromuscular regeneration. *Nat. Commun.* **11**, 1–14 (2020).
- [6] Chen, S. et al. Mechanically and biologically skin-like elastomers for bio-integrated electronics. *Nat. Commun.* **11**, 1–8 (2020).
- [7] Held, M. et al. Soft Electronic Platforms Combining Elastomeric Stretchability and Biodegradability. *Adv. Sustain. Syst.* **6**, 2100035 (2022).
- [8] Yang, Q. et al. Photocurable bioresorbable adhesives as functional interfaces between flexible bioelectronic devices and soft biological tissues. *Nat. Mater.* **20**, 1559-1570 (2021).

We have modified the following sentence in the manuscript.

“..... re-discover the values in unexplored forms beyond conventional, restricted uses as porous scaffolds.”

To

“..... re-discover the values in unexplored forms beyond conventional, restricted uses as porous scaffolds and a few implementations as a substrate for electronic materials [1-3].”

- [1] Cheng, S. et al. Electronic Blood Vessel. *Matter* **3**, 1664-1684 (2020).
- [2] Wang, Q. et al. Shape memory performances of homogeneous poly (L-lactide-co- ϵ -caprolactone)/polytrimethylene carbonate-grafted functionalized graphene oxide nanocomposites. *Eur. Polym. J.* **173**, 111291 (2022).
- [3] Lu, S. et al. Polydopamine-Decorated PLCL Conduit to Induce Synergetic Effect of Electrical Stimulation and Topological Morphology for Peripheral Nerve Regeneration. *Small Methods* (2023) doi.org/10.1002/smt.202200883.

Comment #2: In figure 2, the authors highlight the improvement of PLCL elastomers compared to other works. There likely are other biodegradable elastomers with similar modulus (2e) that can also be stretched to 1000%. This graph also shows hysteresis at 40%, which is exaggerated at higher strains (SI Fig 5), and plastic deformation-- thank you for including this additional characterization.

Our response: We agree with the reviewer. During the cyclic tests at 40 % strain in Figure 2e, the tensile stress decreased by ~5 % and the residual strain increased to ~6 %, while this mechanical property deteriorated at high strains far beyond the elastic region of PLCL (Mn, 200k). Such a tendency appears even more clear in the case of low molecular weight. We expect that this issue could be reduced to some extent by synthesizing PLCL elastomers with high molecular weight, however we need to figure it out through future studies with appropriate chemistry. As mentioned above, we summarized more examples of biodegradable elastomers with their mechanical properties in a supplementary table.

Our modifications: We added the following sentence in the manuscript.

“..... in recent reports. Although the cyclic tests shows hysteresis of PLCL, particularly at high strains, further studies need to improve the property through various chemistry. Furthermore, excellent”

Comment #3: In figure 3g, the authors compare their work with other reported literature. Notably, their work has comparable conductivity to other literature but can stretch to ~600% instead of 300% (references 41 and 42). While this is impressive, it is unclear what the application for this is, and whether this increase is necessary. This is highlighted by the types of device demonstrations that do not show why 600% is better than 300%. Some comments on this point in the manuscript would be appreciated.

Our response: We thank the reviewer for this comment. Just for correction, the stretchability in references 41 and 42 was around 50 %, not 300 %.

Stretchability in conductive polymer composites indicates the maximum strain at which a composite can respond in a reversible way without electrical/mechanical malfunctions, which means a composite with high stretchability can perform a much more stable and reliable function within a desired range of strain. In other words, high stretchability, along with excellent conductivity, is favorable for long-term reliable electrical function over the physiological strain range (~50 %) in biomedical and wearable applications. However, previously reported conductive composites (references 41, 42, 43, and 45 in Figure 3g) showed insufficient stretchability (~50 %) for potential applications, and other examples with high elasticity (references 44 and 46 in Figure 3g) exhibited low conductivity (< ~1 S/cm) as a conductor.

Such highly stretchable composites are believed to contribute to research fields of soft robotics that requires time-dynamic and arbitrary deformation modes along with elaborate, sophisticated functions. We have added this perspective to the manuscript.

Our modifications: We have modified the following sentence in the manuscript.

“..... far surpassed those of degradable composites in previous reports.”

To

“..... far surpassed those of degradable composites in previous reports, which facilitates more robust/reliable function in physiological strain range and may contribute to research applications required for elaborate, sophisticated functions under time-dynamic and arbitrary deformation modes.”

REVIEWER COMMENTS

Reviewer #1 (Remarks to the Author):

The manuscript still needs careful revisions to clarify some critical issues:

1. As the authors agree that temperature changes can affect the output resistance and therefore the measurement of strain. Associated discussions should be provided in the manuscript.

2. PEDOT is a conjugated polymer. It is well known that it is highly stable and not soluble in aqueous solutions (E. M. Ali, E. A. B. Kantchev, H. h. Yu, J. Y. Ying, *Macromolecules* 2007, 40, 6025. <https://pubs.acs.org/doi/pdf/10.1021/ma0708949>). That is why PEDOT:PSS is commercially available in the form of water dispersion as colloidal particles.

There is no clear evidence suggest that PEDOT:PSS is biodegradable in physiological environments. It can only be disintegrated without complete chemical breakdown, leaving the active components in the body. The references the authors provided claiming the biodegradation of PEDOT:PSS are not convincing. For example, reference [1] are studies on PEDOT:Tos, which should not be confused with PEDOT:PSS as different dopants can significantly affect conductivity and solubility. reference [2] shows PEDOT:PSS could be degraded by highly concentrated hydrogen peroxide (~2.5 M) with a long period of time, which is not an environment relevant for biological applications. The instability of PEDOT:PSS in reference [3] is mostly focused on delamination issues, no specific dissolution of PEDOT:PSS is mentioned.

The biocompatibility studies of ionic liquid (P14[TFSI]) in the current work is quite preliminary. As the in vivo biodegradation and metabolic pathways of P14[TFSI] is not clear, the biocompatibility and biodegradability of P14[TFSI] require further validation.

Given current available evidence, calling PEDOT:PSS, P14[TFSI], and device containing these materials “biodegradable” or “bioresorbable” is very misleading. The author should use more precise terms such as “disintegrable” or “partial biodegradable”, not to cause confusion. Please refer to *ACS Cent. Sci.* 2018, 4, 337–348 (DOI: 10.1021/acscentsci.7b00595), composites consisting of PEDOT:PSS and biodegradable polymers should be categorized as “partial degradation” or “disintegrable” materials. Careful modification throughout the manuscript is required. As these materials and devices aim to be implanted and metabolized by human body, the authors should be very careful in presenting their results.

Reviewer #2 (Remarks to the Author):

The manuscript shows substantial improvements over the original submission. Most inconsistencies have been clarified through the revision. However, I still have some reservations regarding the following points:

1. The LED array is purely to demonstrate the mechanical/physical properties. The non-degradable nature of the device should be explicitly explained in the main text. Otherwise, readers without sufficient expertise will easily get confused.
2. The biodegradation of PEDOT:PSS is a controversial topic at this stage. Several studies on the degradation behaviors are carried out under special conditions instead of the standard physiological environment. In addition, the reaction byproducts are not necessarily benign to the biological systems. The authors may want to tune down the claim on the degradability of PEDOT:PSS. The encapsulation may be a reasonable strategy to avoid such a controversy.
3. An electronic mesh is employed for cardiac monitoring and stimulation. All devices are functional over the in vivo experimental period, likely due to the slow biodegradation rate of the PLCL. Mo conductor is particularly vulnerable to leaky encapsulation. Some discussions on the degradation rate of the device are therefore highly encouraged.
4. A tiny strain of 0.05 % is experienced with the PLCL/Mo sensor. Such a small strain range allows stable performance over the in vivo experimental period. However, the circumferential strain of the heart should be much larger than this value. Is it because of the serpentine mesh design or the high modulus of Mo? Some analysis and clarifications are certainly required.

Reviewer #3 (Remarks to the Author):

My comments have been addressed. Editor should defer to the concerns of other reviewers.

Reviewer #1 (Remarks to the Author):

The manuscript still needs careful revisions to clarify some critical issues:

Our response: We appreciate the reviewer's valuable comments. We believe that the revised manuscript, through feedback on the reviewer's comments, will qualify for publishing in *Nature Communications*.

Comment #1: As the authors agree that temperature changes can affect the output resistance and therefore the measurement of strain. Associated discussions should be provided in the manuscript.

Our response: We thank to the reviewer's comment. We have added some text to mention the effect of temperature on the sensitivity of strain sensor to the manuscript and provided the related discussions in detail in Supplementary Information.

Our modifications: We have added the following sentence to the manuscript.

“..... appropriate encapsulation strategy (Supplementary Fig.31).”

To

“..... appropriate encapsulation strategy (Supplementary Fig. 31). The effect of temperature on the sensitivity of PLCL/Mo composite appears in Supplementary Note 11 and Fig. 32.”

We have added the following content and figures to the Supplementary Information.

“Supplementary Note 11. Effect of temperature on the sensitivity of PLCL/Mo composite in physiological condition

The resistance of PLCL/Mo (9:1) composite increases as the temperature rises from 35 °C to 39 °C, primarily due to the decrease in conductivity of Mo flakes and the thermal expansion of PLCL matrix (See the Supplementary Fig. 32a). And, the PLCL/Mo composite showed a similar temperature-independent change in resistance ($R - R_0$), at 0.05 % strain generated by cardiac movements, which was estimated by comparison of Supplementary Figs. S15 and S30 (See the Supplementary Fig. 32b). Separation of these variables might be achieved through use of low thermal conductive materials and encapsulation with thermally-insulative materials. If a temperature sensor was additionally integrated into the cardiac jacket, the temperature change caused by heart malfunctions could be monitored and used for calibrating output signals.”

Supplementary Figure 32. (a) Base resistance changes of PLCL/Mo (9:1) as a function of temperature. (b) Resistance changes of PLCL/Mo (9:1) at a strain of 0.05 % in a temperature range of 35 to 39 °C.

Comment #2: PEDOT is a conjugated polymer. It is well known that it is highly stable and not soluble in aqueous solutions (E. M. Ali, E. A. B. Kantchev, H. h. Yu, J. Y. Ying, Macromolecules 2007, 40, 6025. <https://pubs.acs.org/doi/pdf/10.1021/ma0708949>). That is why PEDOT:PSS is commercially available in the form of water dispersion as colloidal particles.

There is no clear evidence suggest that PEDOT:PSS is biodegradable in physiological environments. It can only be disintegrated without complete chemical breakdown, leaving the active components in the body. The references the authors provided claiming the biodegradation of PEDOT:PSS are not convincing. For example, reference [1] are studies on PEDOT:Tos, which should not be confused with PEDOT:PSS as different dopants can significantly affect conductivity and solubility. reference [2] shows PEDOT:PSS could be degraded by highly concentrated hydrogen peroxide (~2.5 M) with a long period of time, which is not an environment relevant for biological applications. The instability of PEDOT:PSS in reference [3] is mostly focused on delamination issues, no specific dissolution of PEDOT:PSS is mentioned.

The biocompatibility studies of ionic liquid (P14[TFSI]) in the current work is quite preliminary. As the in vivo biodegradation and metabolic pathways of P14[TFSI] is not clear, the biocompatibility and biodegradability of P14[TFSI] require further validation.

Given current available evidence, calling PEDOT:PSS, P14[TFSI], and device containing these materials “biodegradable” or “bioresorbable” is very misleading. The author should use more precise terms such as “disintegrable” or “partial biodegradable”, not to cause confusion. Please refer to ACS Cent. Sci. 2018, 4, 337–348 (DOI: 10.1021/acscentsci.7b00595), composites consisting of PEDOT:PSS and biodegradable polymers should be categorized as “partial degradation” or “disintegrable” materials. Careful modification throughout the manuscript is required. As these materials and devices aim to be implanted and metabolized by human body, the authors should be very careful in presenting their results.

Our response: We agree with the reviewer’s comment that some reports we provided are not enough to validate the biodegradation of PEDOT:PSS in physiological

environment and the in vivo biodegradation and metabolic pathways of P14[TFSI]. To avoid any confusion, we have changed the term ‘biodegradable’ and ‘bioresorbable’ to ‘partially biodegradable’, ‘transient’, or ‘disintegrable’ for PLCL/PP composites and cardiac jackets throughout the manuscript and explained the need for further validation of the long-term biocompatibility of PEDOT:PSS and P14[TFSI] in the manuscript.

Our modifications: We have modified the following sentences in the manuscript.

“Demonstrations of soft electronic grippers and bioresorbable, suture-free cardiac jackets could be the cornerstone

To

“Demonstrations of soft electronic grippers and transient, suture-free cardiac jackets could be the cornerstone

“..... bioresorbable, suture-free cardiac jackets with bio-inspired designs

To

“..... transient, suture-free cardiac jackets with bio-inspired designs

“Figure 3a illustrates degradable, conductive elastomeric composite,

To

“Figure 3a illustrates partially biodegradable, conductive elastomeric composite,

“Figure 4a describes a fully biodegradable soft robotic gripper that can perceive physical stimuli

To

“Figure 4a describes a transient soft robotic gripper that can perceive physical stimuli

“..... and the dissolution rate was slower than that of PLCL itself, presumably due to the hydrophobic PEDOT domains and ionic liquid.”

To

“..... and the dissolution rate was slower than that of PLCL itself, presumably due to the water-insoluble PEDOT domains and hydrophobic ionic liquid.”

“Soft, bioresorbable, suture-free cardiac jackets for heart diseases”

To

“Soft, transient, suture-free cardiac jackets for heart diseases”

“Figure 5d presents a set of images of resorbable electronics coupled to the heart

To

“Figure 5d presents a set of images of transient electronics coupled to the heart

“Such long-lasting, invariant performances of this bioresorbable system can address drawbacks

To

“Such long-lasting, invariant performances of this transient system can address drawbacks

“We couldn’t observe substantial physical degradation of the device throughout the implantation period of 8 weeks due to the slow dissolution rate. However, we believe that constituting materials would gradually degrade as PLCL dissolves. Although chronic degradation of PEDOT:PSS in vivo has not been clearly unveiled yet, there are several reports that PEDOT degrades via hydrolysis in aqueous salt solutions, oxidation by hydrogen peroxide, or cellular phagocytosis by macrophage. Therefore, we expect that combination of these reactions in physiological condition could possibly decompose PEDOT:PSS.

To

“We couldn’t observe substantial physical degradation of the device throughout the implantation period of 8 weeks due to the slow dissolution rate. However, as shown in the decrease in sensitivity of PLCL/Mo composite after 2 weeks of implantation in Supplementary Fig. 31, Mo flakes were being degraded by water molecules penetrating the encapsulation layer at a rate of ~20 nm/day [1], without remaining harmful byproducts. Other constituent materials, such as PEDOT:PSS and P14[TFPI], would be gradually disintegrated as the PLCL matrix dissolved, while the long-term biocompatibility of these materials requires further validation.”

[1] Choi, Y., Koo, J., & Rogers, J. A. Inorganic materials for transient electronics in biomedical applications. *MRS Bull.* **45**, 103-112 (2020).

“PLCL/PEDOT:PSS-based biodegradable, conductive polymer composites (PLCL/PP)”

To

“PLCL/PEDOT:PSS-based partially biodegradable, conductive polymer composites (PLCL/PP)”

“Biodegradable electronic actuator Fabrication of biodegradable, smart soft actuator involve

To

“Transient electronic actuator Fabrication of transient, smart soft actuator involve

“Soft, bioresorbable, suture-free cardiac-integrated electronics”

To

“Soft, transient, suture-free cardiac-integrated electronics”

“Figure 3. Different types of dissolvable conductive elastomers with organic and inorganic fillers. (a) Mechanical strain-dependent electrical characteristics of a degradable conductive elastomer”

To

“Figure 3. Different types of transient conductive elastomers with organic and inorganic fillers. (a) Mechanical strain-dependent electrical characteristics of a partially biodegradable conductive elastomer”

“(d) Examples of dissolvable composite (PLCL/PP)-based electronic component”

To

“(d) Examples of partially biodegradable composite (PLCL/PP)-based electronic component”

“(a) Description of a PLCL-based biodegradable robotic gripper”

To

“(a) Description of a PLCL-based transient robotic gripper”

“Figure 5. Implantable, multifunctional, bioresorbable suture-free cardiac jacket for heart diseases. (a) Schematic illustration of a soft, bioresorbable cardiac jacket”

To

“Figure 5. Implantable, multifunctional, transient suture-free cardiac jacket for heart diseases. (a) Schematic illustration of a soft, transient cardiac jacket”

We have modified the following sentences in the Supplementary Information.

“Fabrication of biodegradable, stretchable electronic circuits.”

To

“Fabrication of partially biodegradable, stretchable electronic circuits.”

“Supplementary Note 9. Biodegradable, smart soft actuator”

To

“Supplementary Note 9. Transient, smart soft actuator”

“Supplementary Note 10. Soft, bioresorbable, suture-free cardiac jacket”

To

“Supplementary Note 10. Soft, transient, suture-free cardiac jacket”

Reviewer #2 (Remarks to the Author):

The manuscript shows substantial improvements over the original submission. Most inconsistencies have been clarified through the revision. However, I still have some reservations regarding the following points.

Our response: We appreciate the reviewer's valuable comments. We believe that the revised manuscript addresses all the scientific/technical issues raised from the reviewer and will qualify for publishing in *Nature Communications*.

Comment #1: The LED array is purely to demonstrate the mechanical/physical properties. The non-degradable nature of the device should be explicitly explained in the main text. Otherwise, readers without sufficient expertise will easily get confused.

Our response: We thank to the review's comment. To avoid any misleadingness, we have mentioned the non-degradable nature of the LED array in the manuscript.

Our modifications: We have modified the following sentence in the manuscript.

“Figure 2j shows an Arduino-powered electronic display with an array (8 x 16) of red, green, and blue (RGB) light-emitting diodes (LEDs) on PLCL substrates, manufactured by microlithography and transfer-printing processes.”

To

“Figure 2j shows a partially-degradable Arduino-powered electronic display with an array (8 x 16) of commercially-available, non-degradable red, green, and blue (RGB) light-emitting diodes (LEDs) on dissolvable PLCL substrates, manufactured by microlithography and transfer-printing processes.”

Comment #2: The biodegradation of PEDOT:PSS is a controversial topic at this stage. Several studies on the degradation behaviors are carried out under special conditions instead of the standard physiological environment. In addition, the reaction byproducts are not necessarily benign to the biological systems. The authors may want to tune down the claim on the degradability of PEDOT:PSS. The encapsulation may be a reasonable strategy to avoid such a controversy.

Our response: We agree with the reviewer's comment. Some reports we provided are not enough to validate the biodegradation of PEDOT:PSS in physiological environment. To address a controversy, we have changed the term ‘biodegradable’ and ‘bioresorbable’ to ‘partially biodegradable’, ‘transient’, or ‘disintegrable’ for PLCL/PP composites.

Our modifications: We have modified the following sentences in the manuscript.

“Demonstrations of soft electronic grippers and bioresorbable, suture-free cardiac jackets could be the cornerstone

To

“Demonstrations of soft electronic grippers and transient, suture-free cardiac jackets could be the cornerstone

“..... bioresorbable, suture-free cardiac jackets with bio-inspired designs

To

“..... transient, suture-free cardiac jackets with bio-inspired designs

“Figure 3a illustrates degradable, conductive elastomeric composite,

To

“Figure 3a illustrates partially biodegradable, conductive elastomeric composite,

“Figure 4a describes a fully biodegradable soft robotic gripper that can perceive physical stimuli

To

“Figure 4a describes a transient soft robotic gripper that can perceive physical stimuli

“..... and the dissolution rate was slower than that of PLCL itself, presumably due to the hydrophobic PEDOT domains and ionic liquid.”

To

“..... and the dissolution rate was slower than that of PLCL itself, presumably due to the water-insoluble PEDOT domains and hydrophobic ionic liquid.”

“Soft, bioresorbable, suture-free cardiac jackets for heart diseases”

To

“Soft, transient, suture-free cardiac jackets for heart diseases”

“Figure 5d presents a set of images of resorbable electronics coupled to the heart

To

“Figure 5d presents a set of images of transient electronics coupled to the heart

“Such long-lasting, invariant performances of this bioresorbable system can address drawbacks

To

“Such long-lasting, invariant performances of this transient system can address drawbacks

“We couldn’t observe substantial physical degradation of the device throughout the implantation period of 8 weeks due to the slow dissolution rate. However, we believe

that constituting materials would gradually degrade as PLCL dissolves. Although chronic degradation of PEDOT:PSS in vivo has not been clearly unveiled yet, there are several reports that PEDOT degrades via hydrolysis in aqueous salt solutions, oxidation by hydrogen peroxide, or cellular phagocytosis by macrophage. Therefore, we expect that combination of these reactions in physiological condition could possibly decompose PEDOT:PSS.

To

“We couldn’t observe substantial physical degradation of the device throughout the implantation period of 8 weeks due to the slow dissolution rate. However, as shown in the decrease in sensitivity of PLCL/Mo composite after 2 weeks of implantation in Supplementary Fig. 31, Mo flakes were being degraded by water molecules penetrating the encapsulation layer at a rate of ~20 nm/day [1], without remaining harmful byproducts. Other constituent materials, such as PEDOT:PSS and P14[TFPI], would be gradually disintegrated as the PLCL matrix dissolved, while the long-term biocompatibility of these materials requires further validation.”

[1] Choi, Y., Koo, J., & Rogers, J. A. Inorganic materials for transient electronics in biomedical applications. *MRS Bull.* **45**, 103-112 (2020).

“PLCL/PEDOT:PSS-based biodegradable, conductive polymer composites (PLCL/PP)”

To

“PLCL/PEDOT:PSS-based partially biodegradable, conductive polymer composites (PLCL/PP)”

“Biodegradable electronic actuator Fabrication of biodegradable, smart soft actuator involve

To

“Transient electronic actuator Fabrication of transient, smart soft actuator involve

“Soft, bioresorbable, suture-free cardiac-integrated electronics”

To

“Soft, transient, suture-free cardiac-integrated electronics”

“Figure 3. Different types of dissolvable conductive elastomers with organic and inorganic fillers. (a) Mechanical strain-dependent electrical characteristics of a degradable conductive elastomer

To

“Figure 3. Different types of transient conductive elastomers with organic and inorganic fillers. (a) Mechanical strain-dependent electrical characteristics of a partially biodegradable conductive elastomer

“(d) Examples of dissolvable composite (PLCL/PP)-based electronic component

To

“(d) Examples of partially biodegradable composite (PLCL/PP)-based electronic component

“(a) Description of a PLCL-based biodegradable robotic gripper

To

“(a) Description of a PLCL-based transient robotic gripper

“**Figure 5. Implantable, multifunctional, bioresorbable suture-free cardiac jacket for heart diseases.** (a) Schematic illustration of a soft, bioresorbable cardiac jacket

To

“**Figure 5. Implantable, multifunctional, transient suture-free cardiac jacket for heart diseases.** (a) Schematic illustration of a soft, transient cardiac jacket

We have modified the following sentences in the Supplementary Information.

“Fabrication of biodegradable, stretchable electronic circuits.”

To

“Fabrication of partially biodegradable, stretchable electronic circuits.”

“**Supplementary Note 9. Biodegradable, smart soft actuator**”

To

“**Supplementary Note 9. Transient, smart soft actuator**”

“**Supplementary Note 10. Soft, bioresorbable, suture-free cardiac jacket**”

To

“**Supplementary Note 10. Soft, transient, suture-free cardiac jacket**”

Comment #3: An electronic mesh is employed for cardiac monitoring and stimulation. All devices are functional over the in vivo experimental period, likely due to the slow biodegradation rate of the PLCL. Mo conductor is particularly vulnerable to leaky encapsulation. Some discussions on the degradation rate of the device are therefore highly encouraged.

Our response: We agree with the reviewer’s comment. We have added relevant contents to the manuscript and enhanced encapsulation approaches of PLCL/Mo sensors to the figure caption of the Supplementary Fig. 31.

Our modifications: We have modified the following sentences in the manuscript.

“We couldn’t observe substantial physical degradation of the device throughout the implantation period of 8 weeks due to the slow dissolution rate. However, we believe that constituting materials would gradually degrade as PLCL dissolves. Although chronic degradation of PEDOT:PSS in vivo has not been clearly unveiled yet, there are several reports that PEDOT degrades via hydrolysis in aqueous salt solutions, oxidation by hydrogen peroxide, or cellular phagocytosis by macrophage. Therefore, we expect that combination of these reactions in physiological condition could possibly decompose PEDOT:PSS.

To

“We couldn’t observe substantial physical degradation of the device throughout the implantation period of 8 weeks due to the slow dissolution rate. However, as shown in the decrease in sensitivity of PLCL/Mo composite after 2 weeks of implantation in Supplementary Fig. 31, Mo flakes were being degraded by water molecules penetrating the encapsulation layer at a rate of ~20 nm/day [1], without remaining harmful byproducts. Other constituent materials, such as PEDOT:PSS and P14[TFSI], would be gradually disintegrated as the PLCL matrix dissolved, while the long-term biocompatibility of these materials requires further validation.”

[1] Choi, Y., Koo, J., & Rogers, J. A. Inorganic materials for transient electronics in biomedical applications. *MRS Bull.* **45**, 103-112 (2020).

We have added the following sentence to the caption of Supplementary Fig. 31.

“**Supplementary Figure 31.** Monitoring of myocardial strain with the PLCL/Mo sensor on the implanted device for 4 weeks to predict cardiac mechanics in the early stages of heart diseases. A myocardial infarction, that occurred when blood flow decreases or stops to the coronary artery of the heart, results in damage of the myocardium, particularly in the left ventricle. This damage accompanies fibrosis generation, which increases the stiffness of cardiomyocytes and thus reduces the elasticity during systole/diastole. Therefore, by monitoring amplitude change of output signals detected by strain sensor, we can predict the relative stiffness of post-MI heart (vs. normal heart).”

To

“**Supplementary Figure 31.** Monitoring of myocardial strain with the PLCL/Mo sensor on the implanted device for 4 weeks to predict cardiac mechanics in the early stages of heart diseases. A myocardial infarction, that occurred when blood flow decreases or stops to the coronary artery of the heart, results in damage of the myocardium, particularly in the left ventricle. This damage accompanies fibrosis generation, which increases the stiffness of cardiomyocytes and thus reduces the elasticity during systole/diastole. Therefore, by monitoring amplitude change of output signals detected by strain sensor, we can predict the relative stiffness of post-MI heart (vs. normal heart). Meanwhile, using a high molecular weight PLCL as encapsulation layers can extend the period of stable operation.”

Comment #4: A tiny strain of 0.05 % is experienced with the PLCL/Mo sensor. Such a small strain range allows stable performance over the in vivo experimental period. However, the circumferential strain of the heart should be much larger than this value. Is it because of the serpentine mesh design or the high modulus of Mo? Some analysis and clarifications are certainly required.

Our response: There are two reasons why mechanical cardiac motions did not interfere with detection of small strains by the strain gauges.

1: As the reviewer pointed out, the soft serpentine mesh absorbed most circumferential strains induced by diastolic/systolic functions of the heart and reduced the strains applied to the sensors.

2. The Young's modulus of the strain sensor (~4 MPa) is much higher than that of the heart (diastolic, ~11 KPa; systolic, ~39 KPa [1]). Therefore, the strain applied to the strain sensor during the mechanical movements would be less than 0.4 %, as estimated in the figure below.

[1] Azeloglu, E. U., & Costa, K. D. Cross-bridge cycling gives rise to spatiotemporal heterogeneity of dynamic subcellular mechanics in cardiac myocytes probed with atomic force microscopy. *Am. J. Physiol. Heart Circ. Physiol.* **298**, H853-H860 (2010).

Our modifications: We have modified the following sentence in the Supplementary Information.

“..... early stages of heart diseases. A myocardial infarction”

To

““..... early stages of heart diseases. Despite the high circumferential strain of the heart (~30 % [1]), the strain applied to the sensor was estimated as low as ~0.05 %, due to the serpentine layout and high modulus (~4 MPa). A myocardial infarction”

[1] Voigt, J-U., & Marta, C. 2-and 3-dimensional myocardial strain in cardiac health and disease. *JACC: Cardiovascular Imaging* **12**, 1849-1863 (2019).

REVIEWERS' COMMENTS

Reviewer #1 (Remarks to the Author):

The paper satisfies all my comments.

Reviewer #2 (Remarks to the Author):

The technical issues have been adequately addressed through the revisions. I recommend the manuscript be considered for publication in its current form.